# Inferring protein expression changes from mRNA in Alzheimer's dementia using deep neural networks

Shinya Tasaki [1✉], Jishu Xu[1], Denis R. Avey[1], Lynnaun Johnson[1], Vladislav A. Petyuk [2], Robert J. Dawe[1], David A. Bennett[1], Yanling Wang[1] & Chris Gaiteri[1]

Identifying the molecular systems and proteins that modify the progression of Alzheimer's disease and related dementias (ADRD) is central to drug target selection. However, discordance between mRNA and protein abundance, and the scarcity of proteomic data, has limited our ability to advance candidate targets that are mainly based on gene expression. Therefore, by using a deep neural network that predicts protein abundance from mRNA expression, here we attempt to track the early protein drivers of ADRD. Specifically, by applying the clei2block deep learning model to 1192 brain RNA-seq samples, we identify protein modules and disease-associated expression changes that were not directly observed at the mRNA level. Moreover, pseudo-temporal trajectory inference based on the predicted proteome became more closely correlated with cognitive decline and hippocampal atrophy compared to RNA-based trajectories. This suggests that the predicted changes in protein expression could provide a better molecular representation of ADRD progression. Furthermore, overlaying clinical traits on protein pseudotime trajectory identifies protein modules altered before cognitive impairment. These results demonstrate how our method can be used to identify potential early protein drivers and possible drug targets for treating and/or preventing ADRD.

[1] Rush Alzheimer's Disease Center, Rush University Medical Center, Chicago, IL, USA. [2] Biological Sciences Division, Pacific Northwest National Laboratory, Richland, WA, USA. ✉email: stasaki@gmail.com

Without transformative drug discovery, Alzheimer's disease and related dementias (ADRD) are projected to affect 100 million individuals in 2050[1]. To find molecular drug targets for ADRD, community-based aging cohort studies and patient-centered research consortia have generated thousands of brain transcriptomes from individuals at various stages of disease[2]. However, the large discrepancies between mRNA and protein make it challenging to confidently advance mRNA findings in drug discovery because drugs typically target protein. The severity of this disconnect is illustrated by the fact that the most classic protein signs of ADRD—accumulations of β-amyloid and helical filament tau (PHFtau) pathology—are not accompanied by transcriptional changes of APP and MAPT genes[2], and hence would not be detected by the standard transcriptome analysis. However, proteomic characterization of ADRD brains still lags behind transcriptomics. Therefore, developing a method to more accurately relate mRNA levels to proteins would dramatically increase the utility of existing transcriptome data. The predicted proteomes would inherit the size and regional diversity of RNA-seq, and enable systems biology analysis of ADRD at the protein level.

To accelerate drug discovery via predicted protein levels, we perform a deep-learning-based, multi-omic analysis of postmortem brain samples from participants of two large, longitudinal cohort studies of aging and dementia, which have extensive antemortem cognitive assessments, postmortem neuropathologic data, and structural brain imaging[3]. Specifically, we utilize RNA-seq data from the dorsolateral prefrontal cortex (DLPFC) ($n = 1192$) generated through the Accelerating Medicines Partnership-AD (AMP-AD). For a subset of participants ($n = 384$), global proteomics using tandem mass tags (TMT) from the DLPFC was conducted previously[4]. This multi-omics data enables us to build a deep learning model to predict differences in protein abundance between individuals from RNA-seq data.

We conduct a series of systems biology analyses to evaluate whether the predicted proteome can be utilized to identify novel drug targets and molecular mechanisms of ADRD. In the first application, we perform a molecular coexpression approach to defining molecular systems and disease-associated changes[5]. The basis for this analysis is to identify covaried proteins across individuals in a cohort. The groups of covaried proteins termed "modules" represent the activity of major molecular systems.

In the second application, we test the accuracy of predicted differential protein abundance in ADRD. In particular, we examine the predicted protein levels for genes harboring noncoding variants associated with AD[6,7]. Since the massive AD GWAS (genome-wide association study) efforts ostensibly direct drug development efforts towards particular mechanisms, the lack of mRNA effects for these GWAS genes has been limiting.

The third major application of this model addressed a fundamental challenge to cross-sectional brain omic studies: differentiating upstream molecular changes from downstream or correlative changes in ADRD. Pseudo-temporal trajectory analyses - which are often utilized in single-cell transcriptomics[8] - are responsive to major trends in unlabeled samples, which may be related to disease progression[9,10]. Here, we employ a pseudo-temporal reconstruction approach based on estimated proteomes, as well as RNA-seq data.

Due to the integrative nature of deep neural networks, we find that it is possible to robustly predict protein levels, despite their divergence from corresponding mRNA. Further, we find comparable accuracy of this model in predicting protein levels for an entirely separate cohort. The systems biology applications of deep neural networks to predictive protein abundance indicate that it can blend the advantages of the wide availability of transcriptomics with the strong phenotypic relationships of protein, to enable a more accessible and disease-relevant target selection for drug discovery.

## Results

**Deep-neural protein translation of brain transcriptome**. The transcriptome, proteome, and phenotypic data utilized in this study originate from donors in two longitudinal, community-based cohort studies of aging and dementia—the Religious Orders Study (ROS) and the Rush Memory and Aging Project (MAP), referred to as ROSMAP[3]. Collectively, ROSMAP has enrolled >3500 older persons, all of whom have agreed to brain donation and annual clinical and detailed cognitive evaluation. Brain transcriptomes of 1192 individuals and unbiased global proteomes of 384 individuals in the DLPFC brain region (Table S1) were profiled by our group and others as part of the AMP-AD (syn3388564 and syn17015098).

Out of 8391 proteins measured in ROSMAP, only 3096 proteins showed a minimally significant concordance with corresponding gene expression (false discovery rate (FDR) at 5%, $n = 384$), and the average correlation for all proteins was 0.09. This low correlation is indicative of the high degree of discordance between RNA and protein, which was also observed in an independent ADRD brain cohort from the Mount Sinai brain bank (MSBB) (mean $R = 0.11$, $n = 196$). Thus, even with large ($n = 100+$) sample sizes, standard transcriptome analysis will likely miss many proteins truly related to ADRD, challenging a premise of transcriptome analysis, wherein RNA expression levels are a surrogate for protein abundance. This is indeed the case with the ROSMAP cohort: considering the 384 ROSMAP brain samples wherein both mRNA and protein are available, differentially expressed genes (65, Bonferroni-corrected $p < 0.05$) and differentially abundant proteins (201) for AD diagnosis only overlap in two instances (one-sided Fisher's exact test, $p$-value = 0.51). To overcome the discrepancy of protein levels and corresponding mRNA levels, we developed a predictive model of the proteome based on multiple variables extracted from transcriptome data. The application of this model to large-scale brain transcriptomes allowed us to predict estimated protein abundance and conduct subsequent systems biology analyses to identify early influential changes in specific protein levels (Fig. 1).

The central hypothesis underlying the model is that protein abundance is determined not only by the abundance of mRNA coding the protein itself but also by other aspects of the tissue state, which are themselves diffusely represented in the transcriptome profile of the tissue. To estimate the tissue state that can inform the global proteome profile, we modified the scGen framework[11] that is designed to find the latent state representation of the transcriptome profile. Specifically, each transcriptome profile is encoded into low dimensional probabilistic distributions and then the decoder network takes an encoded vector sampled from the distributions to create the corresponding proteome abundance. Then, this decoded proteome was merged with the transcript features of each protein via a linear regression (LR) layer to generate a predicted proteome profile (Fig. 1). We designed this "clei2block" model as an ensemble of deep neural models with 12 combinations of predictive features from mature-RNA, pre-mRNA, and the 3′ untranslated region (UTR) length (Figs. S1 and S2). We used out-of-fold predictions to evaluate the performance of the model. Specifically, we split the samples into ten folds, taking care to balance cognitive status and brain pathologies. Then, for each fold, we removed the fold from the data and used the remaining data to train the model. We repeated this process for each of ten holds, resulting in obtaining 10 models trained with different training samples and holdout testing data.

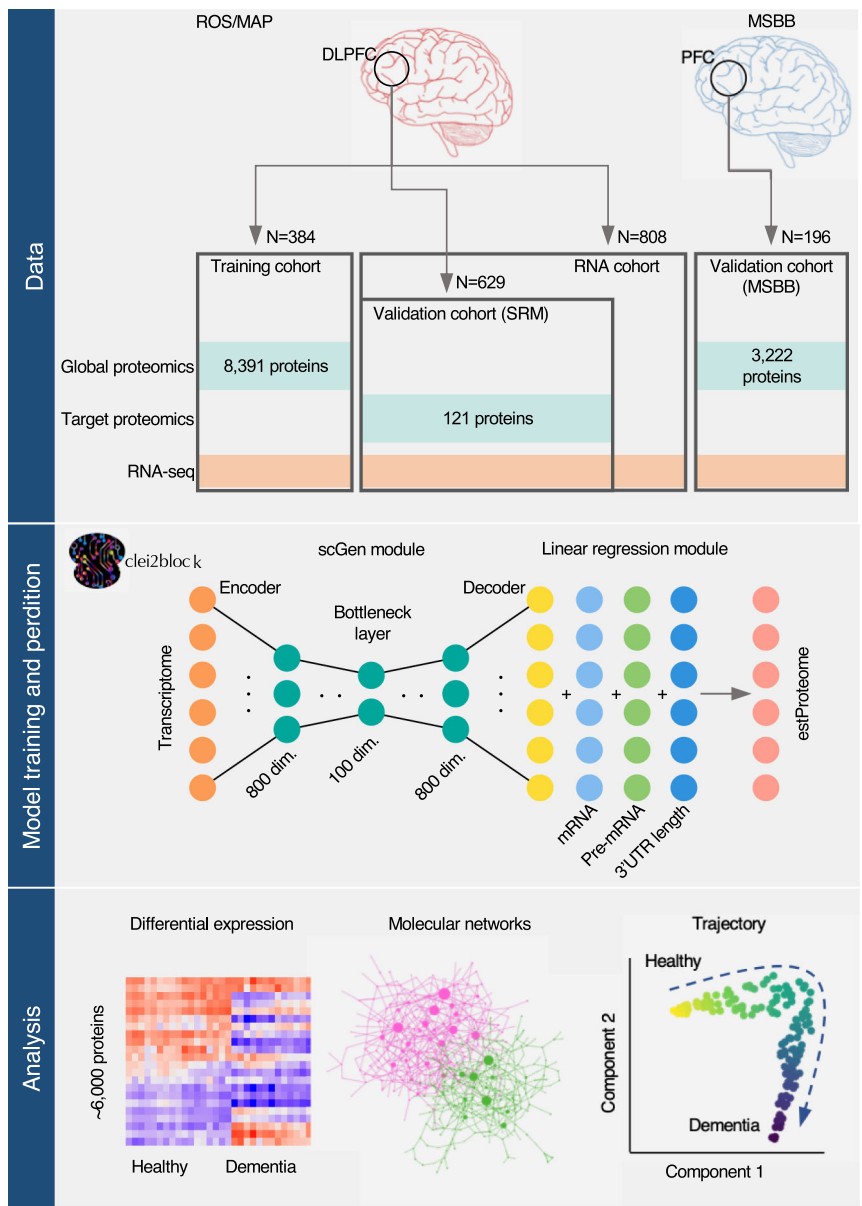

**Fig. 1 Deep-neural protein translation enables system-wide proteomic analysis in older brains.** We built a predictive model called clei2block to estimate protein abundance from RNA expression and applied the model to the large-scale brain transcriptome data from older adults. We showed the utility of estimated proteomic data to understand ADRD molecular pathogenesis via a range of systems biology approaches.

To calculate overall prediction accuracy, predictions for the holdout samples were concatenated (Fig. S3).

The consensus prediction of the clei2block model showed significant positive correlations with the actual abundance of 5998 proteins (FDR 5%), and the average correlation for all proteins was 0.18 (Supplementary Data 1), which is an approximately 2-fold increase compared to the raw RNA expression (Fig. 2a). Also, our model outperformed two representative baseline methods, linear-regression-based (Elastic-net) and boosting tree-based (CatBoost) models (Fig. 2b).

To examine if the sample size ($n = 384$) is sufficient for building a model predicting protein abundance, we trained the clei2block model with sub-sampled training data. The result indicates the predictive accuracy is saturated and the performance gain from an increased sample size will be minimal (Fig. S4).

To determine which aspects of the clei2block are chiefly responsible for its accuracy, we compared three models: (i)

clei2block that is a combination of the scGen and LR modules, (ii) the NN-linear that is a combination of a fully connected layer and the LR module, (iii) the scGen model. The clei2block model (i) performed the best, followed by the scGen and then the NN-linear model (Fig. 2c). This indicates that the scGen module that operates latent-space encoding with non-linear transformation is critical to achieving higher predictive accuracy.

To validate the prediction, we utilized targeted proteomics data of 121 proteins using selected reaction monitoring (SRM) for 637 samples independent of the training sample. The model showed superior performance for targeted proteomics data over the raw RNA expression (Fig. 2d). We also applied the model to brain transcriptome data from the MSBB cohort and compared the estimated proteome with global proteomics data ($n = 196$). This is a challenging assessment, as tissue sampling, brain region, sample preparation, measurements, and data processing were conducted independently from the ROSMAP cohort. However,

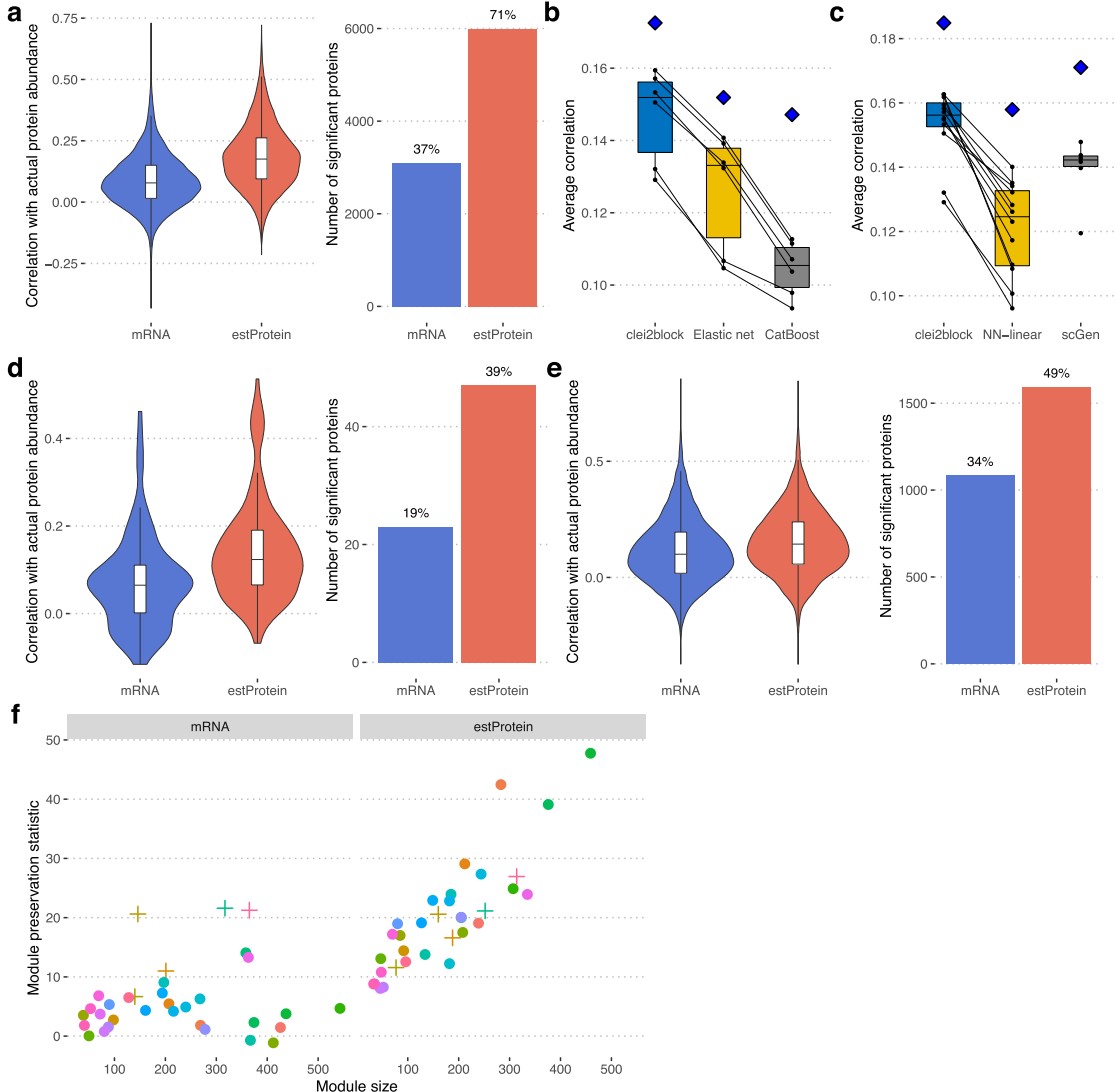

the predicted proteomes based on the model trained with ROSMAP data showed better concordance with actual proteins than did the raw RNA expression from that very cohort (Fig. 2e). This result indicates that the predictive model can be generalized to other cohorts or protein measurement methods.

**Predicted proteomes gain realistic network properties**. Coexpression network analysis is a common approach to discovering gene or protein systems associated with the disease. Therefore, we compared the coexpression network structure between proteomes, transcriptomes, estimated proteomes. For these analyses, we used high-quality predictions for the 5998 proteins (FDR 5%, $R > 0.1$) and 808 samples from the DLPFC (RNA cohort) that were never used in the training process, ensuring unbiased and rigorous assessments.

The coexpression network structure between proteomes and transcriptomes is quite different, with only 6 of 33 protein modules preserved, mostly cell-type-related modules (Fig. 2f). However, when the same transcriptomes were transformed into estimated proteomes, the network structures of 29 modules became highly concordant with the actual proteomes (Fig. 2f). This result demonstrates that, in addition to aligning individual proteins, the estimated levels facilitate detecting system-level behavior of brain proteomes.

**Influential predictors for protein abundance**. To track down the origin of predictive accuracy, we conducted an in-depth investigation on the predictive model. First, we evaluated the collective contribution of each data type to the prediction accuracy. We trained clei2block with the removal of each data type and compared the performance with the model trained with a full dataset. This evaluation quantifies the predictive power uniquely attributed to each data type. Removing any of the data types reduced the predictive performance (Fig. 3a), suggesting that blending different types of data is beneficial for predicting protein abundance. Particularly we observed the largest performance drop with the lack of mRNA expression in the linear-regression module followed by 3′UTR and mRNA inputs in the scGen modules.

Next, to dissect the variables, rather than the data types, that contributed to protein prediction, we calculated the SHAP (Shapley Additive exPlanations) score for each input of our predictive model. For this analysis, we used the model using principal components (PCs) of mRNAs as an encoder input and mRNA, pre-mRNA, and the 3′ UTR length as inputs for the LR module. Overall, mRNA levels of the corresponding protein are the most informative predictor, reiterating the central dogma of molecular biology. Several PCs showed greater or comparable contributions than those of the protein-specific inputs. Interestingly, genes projecting to influential PCs were enriched for cell-

**Fig. 2 Deep neural networks improve protein–RNA concordance. a** Test data performance of the clei2block model. The violin plot is based on Pearson's correlation between actual and predicted protein abundance or its corresponding mRNA level. To generate this plot, 7925 and 8391 different genes were used for mRNA and estProtein, respectively. The upper, center, and lower line of the boxplot indicates 75%, 50%, and 25% quantile, respectively. The upper and lower whisker of the boxplot indicates 75% quantile +1.5 * interquartile range (IQR) and 25% quantile −1.5 * IQR. The bar plot indicates the number of mRNAs and estimated proteins positively correlated with actual protein abundance at FDR 5%. The percentage of the concordant proteins out of all measured proteins was displayed above the bar. **b** Performance comparison of clei2block with Elastic-net and CatBoost. We trained the Elastic-net and CatBoost models for each protein separately using the same data and sample splits with those of the clei2block model. Because training the Elastic-net model took a significantly long time with a large number of predictors, we focused on the six submodels that take mRNA-PCs, pre-mRNA-PCs, or UTR-PCs as predictors in this comparison. The boxplot represents the performance of these six models for all groups. A blue diamond indicates the performance of ensemble prediction. The upper, center, and lower line of the boxplot indicates 75%, 50%, and 25% quantile, respectively. The upper and lower whisker of the boxplot indicates 75% quantile +1.5 * interquartile range (IQR) and 25% quantile −1.5 * IQR. **c** Performance comparison of different neural net architectures. The clei2block was compared with the model without the scGen component and the scGen models. To generate this plot, the performance of 12, 12, and 6 models were used for clei2block, NN-linear, and scGen, respectively. A blue diamond indicates the performance of ensemble prediction. The upper, center, and lower line of the boxplot indicates 75%, 50%, and 25% quantile, respectively. The upper and lower whisker of the boxplot indicates 75% quantile +1.5 * interquartile range (IQR) and 25% quantile −1.5 * IQR. **d** Validation of model performance in SRM cohort. To generate this plot, 121 different genes were used for both mRNA and estProtein. The upper, center, and lower line of the boxplot indicates 75%, 50%, and 25% quantile, respectively. The upper and lower whisker of the boxplot indicates 75% quantile +1.5 * interquartile range (IQR) and 25% quantile −1.5 * IQR. **e** Validation of model performance in MSBB cohort. To generate this plot, 3254 and 3305 different genes were used for mRNA and estProtein, respectively. The upper, center, and lower line of the boxplot indicates 75%, 50%, and 25% quantile, respectively. The upper and lower whisker of the boxplot indicates 75% quantile +1.5 * interquartile range (IQR) and 25% quantile −1.5 * IQR. **f** Congruence of co-abundance module structures between actual proteome and estimated proteome. Protein modules were created based on the actual proteome data with SpeakEasy. To examine whether protein modules are preserved in transcriptome or estimated proteome, we ran the modulePreservation function implemented in the WGCNA R package. For the transcriptome data, we used the data from the same individuals where the actual proteome data was measured. For the estimated proteome data we used the estimated data from independent individuals without proteome measurements for rigorous assessment of module preservation. Enrichment of brain cell-specific genes for proteomic modules was evaluated based on Fisher's exact test and the modules that passed the Bonferroni-corrected p-value less than 0.05 were indicated as + symbol.

type-related genes such as immune response and neurogenesis, and protein translational and degradation, such as ribosomal proteins and a proteasome accessory complex (Fig. 3b and Supplementary Data 2). The results suggested that cell composition, as well as mRNA levels of translational and post-translational machinery that regulate protein abundance, provided critical information to improve the prediction accuracy.

**Molecular characteristics define protein predictability from RNA-seq.** It is important to understand which kinds of proteins can or cannot be predicted by the model. We conducted an extensive investigation in protein characteristics defining its predictability by RNA-seq (Fig. 3c and Supplementary Data 3). We found that the proteins with higher mean intensity in the TMT measurement showed improvement in predictive performance by the model, more so than those with lower mean intensity. Conversely, the proteins with higher variance in the TMT measurement had less benefit from the model, as those proteins already have a higher correlation with their corresponding mRNA levels. In addition, we explored various system-level molecular characteristics. We found that connectivity in protein co-abundance networks and protein–protein interaction networks, the number of transcription factors bound to the promoter region, and the number of RNA-binding proteins bound to mRNA adjusted with the length of mRNA were all positively correlated with the magnitude of performance improvement by the predictive model (Fig. 3c and Supplementary Data 3). Together these molecular characteristics explained 14% of the improvement in the predictive accuracy. Overall, our model performed well for the genes whose proteins have relationships with many other proteins, consistent with the logic that such proteins likely have broad latent representation in RNA data.

**Predicted proteomes can facilitate the discovery of AD-related proteins.** Next, we examined whether the predicted proteomes gain ADRD-related signals that are not seen in transcriptomes.

Results of this examination will determine to what extent the estimated protein data can improve our understanding of ADRD. To do this, we conducted differential expression analysis between AD and control cases, based on (1) the transcriptomes, (2) the actual proteomes, and (3) the estimated proteomes, and then contrasted test statistics of the differential expression. To ensure no bias from the training samples, we used the RNA cohort for #1 and #3 and the TMT cohort for #2. The correlation of differential expression statistics of actual proteomes and transcriptomes are moderate ($R = 0.38$) (Fig. 4a and Supplementary Data 4), which again indicates the limitations of transcriptome data in finding AD-related proteins. However, differential expression statistics of the estimated proteomes were highly concordant with those of the actual proteomes ($R = 0.88$), in contrast to the baseline transcriptome vs. estimated-proteome relationship ($R = 0.39$). Moreover, the association of predicted protein modules with AD diagnosis is highly correlated ($R = 0.91$) with that computed from actual proteome data (Fig. S5). We stratified the proteins based on the test set accuracy and examined how the accuracy impacted the detection of disease signals. The result indicates that even for proteins with lower correlation, their disease effects are in line with the actual observation better than mRNA (Fig. 4b). To verify the result with independent data sets, we repeated the same analysis for the SRM cohort and the MSBB cohort. For both data sets, the consistency between actual protein and mRNA was improved by converting mRNA into estimated protein using the clei2block model trained with the ROSMAP TMT data set (Fig. 4c–f and Supplementary Data 4).

To further provide more evidence that our model indeed facilitates disease protein discovery, we conducted an additional analysis with the MSBB data. We followed the typical scenario for discovering disease-related proteins with RNA-seq data, where (i) run association analysis with the RNA-seq data, (ii) select candidates from differentially expressed genes, (ii) measure protein levels for those genes to validate that they are differentially expressed at the protein level. We used the CERAD (Consortium to Establish a Registry for Alzheimer's Disease) score, neuropathologic

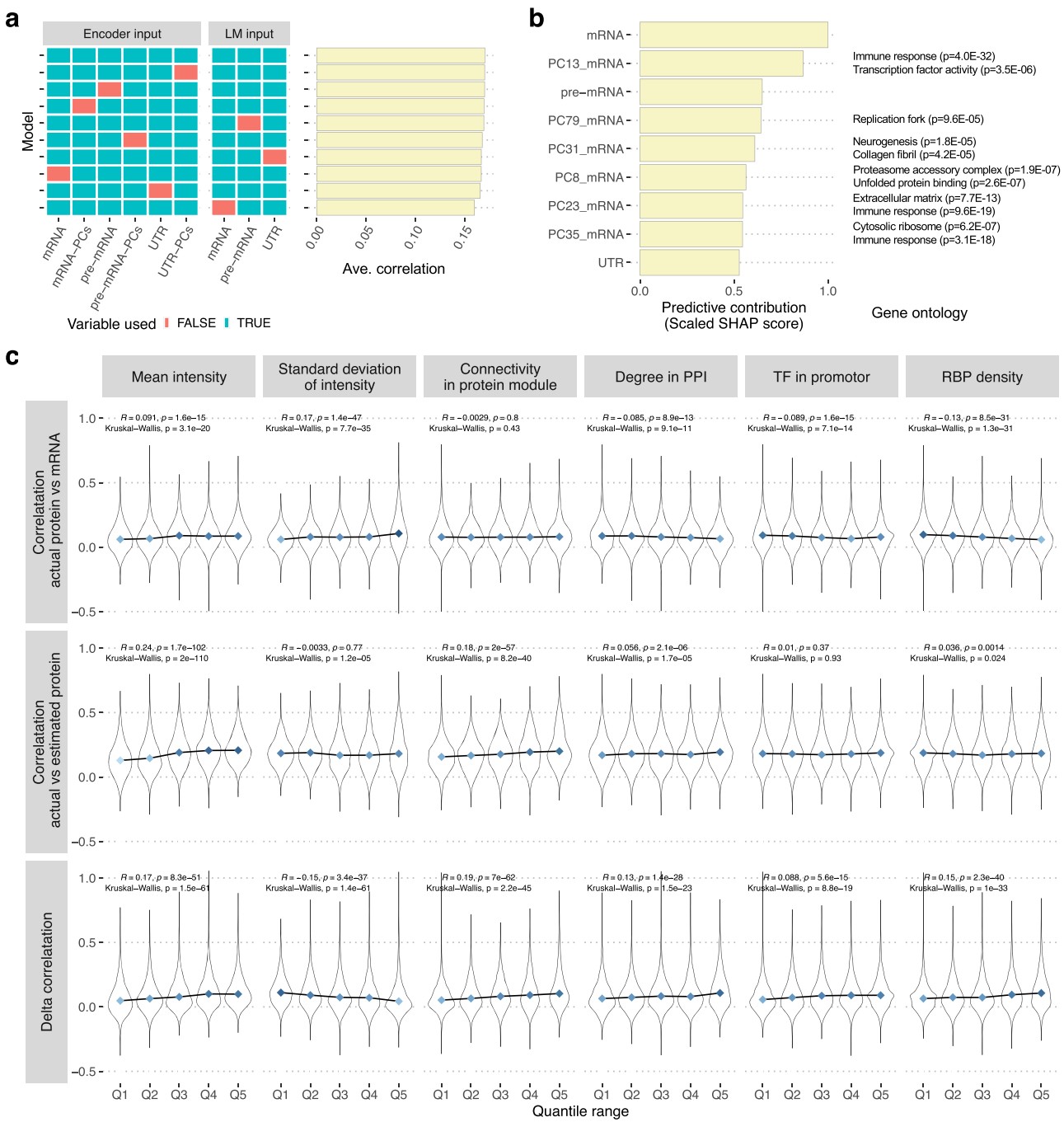

**Fig. 3 Understanding molecular basis of protein prediction. a** Ablation experiment to identify key data types. The clei2block model was trained with the removal of entire input variables of each data type as indicated in the heatmap. The average Pearson's correlations between predicted and actual protein levels for each model are indicated in the bar graph on the right. **b** Influential predictors for protein abundance. The predictive contribution of each variable in the model with mRNA-PCs and all LM inputs was estimated using GradientExplainer. Representative gene ontologies enriched for genes contributing to the influential PCs were described for each PC. The enrichment analysis was conducted using one-sided Fisher's exact test. The significance levels were adjusted for multiple comparisons using FDR at 5%. **c** Molecular characteristics defining protein predictability. Correlations of protein with mRNA and estimated protein and the difference between these were compared with characteristics in the protein measurement and molecular interactions.

diagnosis of AD, as a phenotype for this experiment. We picked the top N-genes associated with the phenotype in RNA-seq data and the estimated protein data, respectively. Then we examined how many of them are differentially expressed with the actual protein data from the same individuals (Fig. 4g). For instance, out of the top 100 significant genes in RNA data, 69 matched the direction of the association at the actual protein levels, and of these 23 showed p-value < 0.05. In contrast, 80 out of the top 100 genes in the

predicted protein data matched the direction of the association with the actual protein data and 36 showed p-value < 0.05. For reference, we also selected genes randomly, resulting in an 11% success rate. Our prediction model improved the success probability of disease protein discovery 1.6-fold over the RNA-based gene selection and 3.4-fold over random selection. This demonstrates that the clei2block deep learning approach significantly improves disease protein discovery over traditional processes.

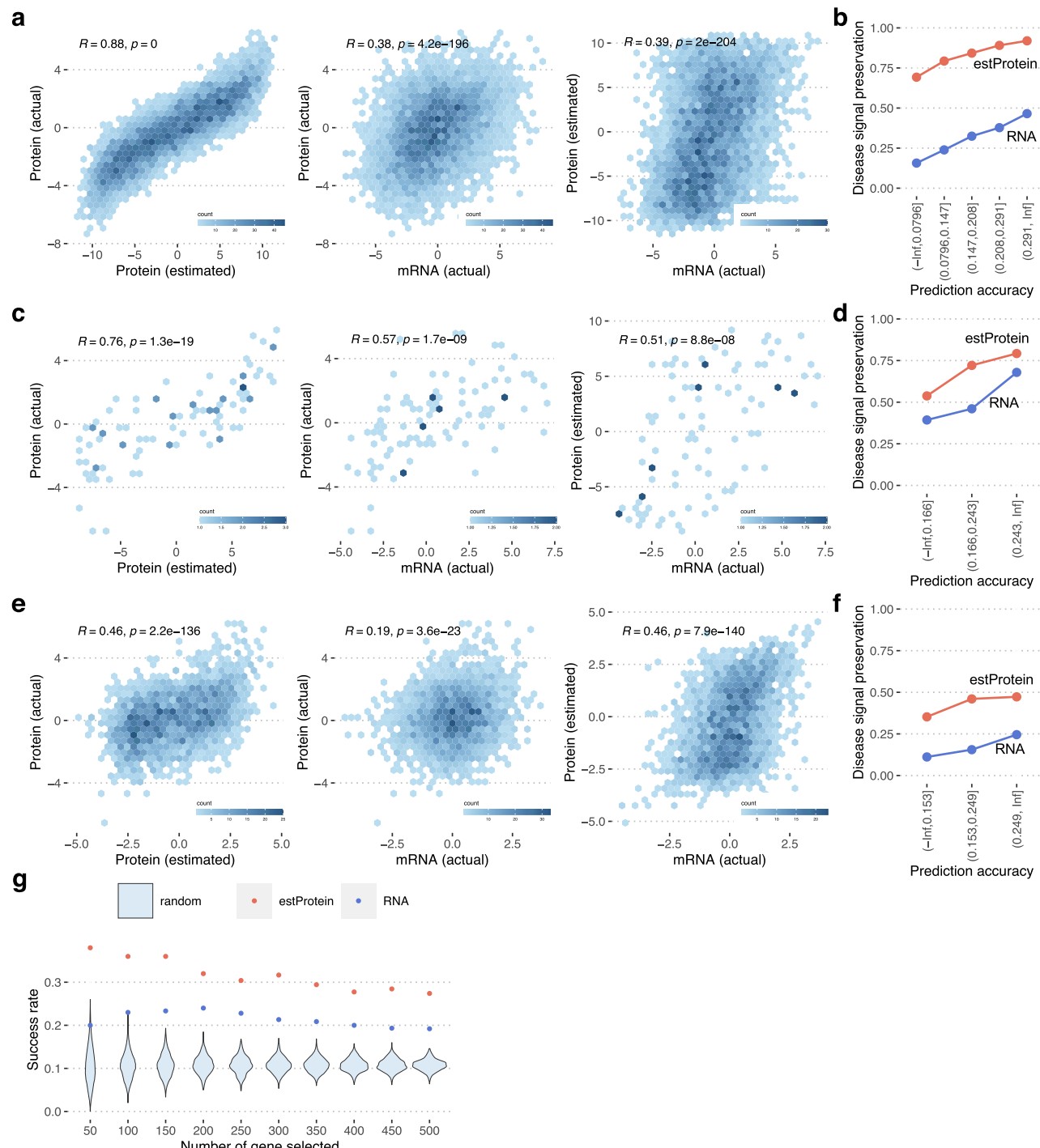

**Fig. 4 Differential expression statistics of the estimated proteomes. a** Estimated proteome greatly improved the concordance of AD-related signals between RNA-seq data and protein data. We conducted case–control studies to find differentially expressed genes in AD based on transcriptome data, the actual TMT proteome data, and estimated proteome data. To maximize the similarity of AD-associated genes, we used the same individuals for the transcriptome data and the actual proteome data. Whereas, to prevent any information leakage from the actual proteome data, we used estimated proteome data of an independent subset. T-statistics for AD diagnosis were used for this comparison. We used Pearson's correlation for the comparison. **b** Pearson's correlation between predicted and actual AD association in the TMT proteome data stratified by prediction accuracy. Proteins are stratified based on the prediction accuracy of the clei2block model in testing data. For each group of genes, we compared t-statistic for AD diagnosis. We repeated the same analysis for (**c, d**) the SRM cohort and (**e, f**) the MSBB cohort. We used Pearson's correlation for these comparisons. The source of prediction accuracy used in the stratification analyses is the same as one for Fig. 4b, but we stratified proteins into three groups instead of four due to the smaller number of proteins in the SRM and MSBB data. **g** Estimated proteome accurately identifies differentially expressed proteins in AD. The clei2block model trained with ROSMAP data was applied to MSBB data, which is entirely independent of the ROSMAP cohort. AD-related genes identified from the estimated protein data are more likely validated with the actual proteome data than those from raw RNA expression.

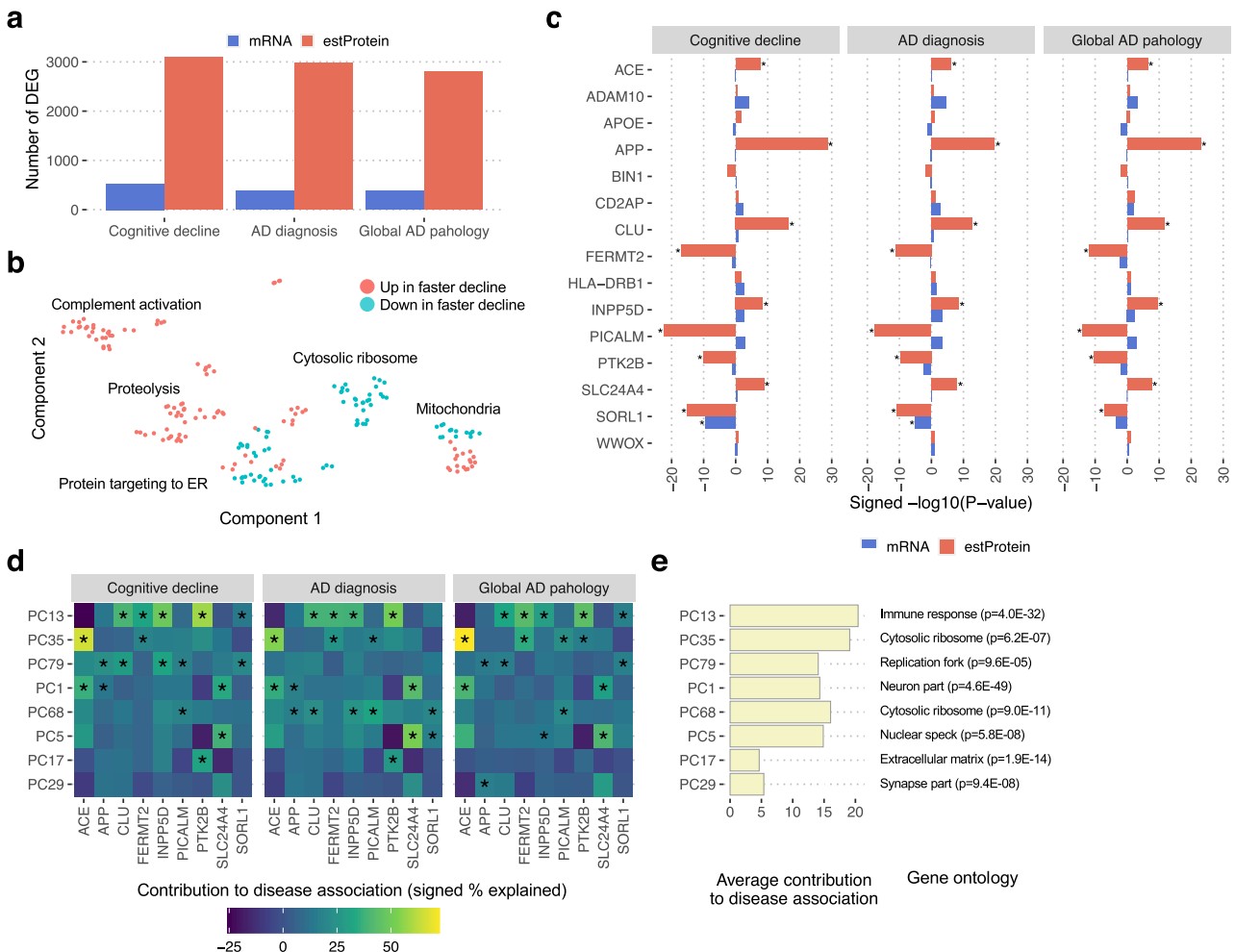

**Fig. 5 Estimated proteome identifies proteins associated with the cognitive decline with greater significance than does the transcriptome. a** The number of genes associated with cognitive decline and other ADRD-related traits. Estimated proteomes and transcriptomes from the same individual (RNA cohort, $n = 808$) were tested for their associations with traits using limma. A Bonferroni-corrected *p*-value less than 0.05 was set as a significance threshold. **b** GO enrichment map for protein-specific cognitive-decline associated genes. One-sided Fisher's exact test was conducted to assess GO enrichment in genes significantly associated with cognitive decline at the protein level but not the mRNA level. The significant GOs at FDR 5% were visualized as a 2D map embedded by UMAP. **c** Differential expression statistics of AD GWAS genes at the protein and the mRNA level. The test was conducted using limma. We used the same significance threshold with Fig. 5a. **d** Influential predictors for the protein-trait association. The predictive contribution of each variable in the encoder inputs of mRNA-PCs was estimated using GradientExplainer. For each protein-trait pair, we selected the top two PCs, indicated by an asterisk, that strengthened their association. The color of the heatmap represents the contribution of each PC to the protein variance explained by trait in percentage. **e** Overall contributions of PCs to the associations across AD GWAS proteins and the traits. Representative gene ontologies enriched for genes contributing to the influential PCs were described in the bar plot. The enrichment analysis was conducted using one-sided Fisher's exact test. The significance levels were adjusted for multiple comparisons using FDR at 5%.

**Proteins associated with cognitive decline.** Slowing down or halting the rate of cognitive decline is the primary clinical endpoint for any dementia therapy. Therefore, to examine whether estimated proteomics can identify proteins related to cognitive decline, we conducted a proteome-wide association analysis against the person-specific slope of the cognitive decline trajectory based on annual cognitive evaluations[12]. In this assessment, we used the 808 samples of high-quality predictions for 5731 genes measured in both RNA-seq and TMT proteomics. We found 3277 proteins associated with cognitive decline (Bonferroni-corrected $p < 0.05$) (Fig. 5a and Supplementary Data 5), which is 13-fold more than those identified from the actual proteome data of 384 samples, and 6-fold more than those from transcriptome data of the same 808 samples. The greater numbers of differentially expressed genes in estimated proteomes were also seen for AD diagnosis, global AD pathology (Fig. 5a), and

cognitive score (Fig. S6). There are substantial similarities in the ADRD-associated proteins between actual and estimated data across different traits, but they are clearly distinct from mRNA results (Fig. S7).

Next, we applied the model to RNA-seq data from the posterior cingulate gyrus ($n = 645$) and the anterior caudate ($n = 705$) regions from the ROSMAP cohort to examine whether the model trained by DLPFC samples can also augment the connection of RNA-seq data with cognitive decline in different brain regions. Indeed, we detected a notable increase in the number of genes associated with cognitive decline based on the estimated proteome compared to the mRNA (Fig. S8).

To reveal biological pathways of protein-specific differentially expressed genes, we conducted gene ontology (GO) analysis for genes associated with cognitive decline at the protein level (Bonferroni-corrected $p < 0.05$) but not at the mRNA level

(nominal $p > 0.05$). There is strong biological enrichment in elevated immune response, proteolysis, and a part of mitochondrial pathways for faster cognitive decline, whereas higher ribosomal proteins and endoplasmic reticulum proteins, and some mitochondrial protein levels were associated with slower cognitive decline (Fig. 5b and Supplementary Data 6).

**AD GWAS associated with cognitive decline**. Understanding how genetic susceptibility genes for AD may drive cognitive decline is an important and largely unresolved question for ADRD pathogenesis. To this end, we focused on 15 AD-GWAS genes[6] with both mRNA and estimated protein data. The direction of disease-related changes was overall concordant between mRNA and estimated proteins, but proteins showed a stronger association with cognitive decline (Fig. 5c). Notably, APP and PICALM proteins exhibited several orders of magnitude greater associations with cognition than their mRNAs. Specifically, higher APP predicted protein explained 18.9% of faster cognitive decline, whereas APP mRNA levels showed weak and even reversed association (Fig. S9). For PICALM, lower protein levels explained 14.3% of faster cognitive decline, in line with the previous reports showing the decreased PICALM protein in AD[13].

We conducted an in-depth analysis to determine what biological systems contribute to the associations of AD genes with AD diagnosis, cognitive decline, and global AD pathology. Specifically, we removed the contribution of each PC in mRNA-PCs from the estimated protein levels and conducted differential expression analysis. Then we compared the variance of protein levels explained by the trait with that of the original estimation (Fig. 5d). For instance, by removing the effect of the 1st PCs of mRNAs (PC1), the variance of APP explained by cognitive decline was reduced by 13.5%. PC1 was strongly associated with neuronal genes. This might indicate that APP protein is predicted through the number of neurons or molecular state changes in the neuronal population. We averaged the contributions across AD genes and traits and found that immune gene signature affected the associations the most, followed by ribosomal proteins, nuclear speckles, and neuronal genes overall (Fig. 5e).

**Protein-state trajectory reflects ADRD progression**. Given insightful findings from trajectory analyses for ADRD using bulk RNA-seq data, we hypothesized that dementia progression can be modeled as state transitions in the brain proteome. To test this hypothesis, we applied a top-performing algorithm for linear trajectory identification[14] to the predicted proteomes of 1192 samples from DLPFC and inferred the protein-state trajectory among diverse older adults (Fig. 6a). We note that the trajectory was estimated purely based on estimated protein levels without any guidance from clinical or pathologic phenotypes or any prior protein selection. The protein pseudotime progression was associated with AD diagnosis (Fig. 6a) and global AD pathology (Fig. 6b), which contrasted with mRNA pseudotime, only showing moderate association with these traits (Fig. 6a, b).

Notably, the protein pseudotime was associated with cognitive decline ($R = -0.34$, $p = 4.4e-32$) stronger than the RNA pseudotime ($R = -0.13$, $p = 1.6e-5$) (Fig. 6c). The associations with cognition score also showed the same trend as that of cognitive decline (Fig. S10). Variance decomposition analysis indicated that the protein pseudotime explained 11.2% of cognitive decline independent of age at death (var. exp., 3.3%) and the RNA pseudotime (var. exp.,1%). This suggests that the protein pseudotime based on the predicted proteomes captures a significant component of cognitive decline, thus representing the effectiveness of our computational protein translation approach in understanding ADRD progression.

To track down the potential origin of association between pseudotime and cognitive decline, we compared each sample's position in pseudotime with hemisphere volume ($n = 459$), as well as hippocampus volume, a key brain region for memory and learning. We found that individuals with advanced protein pseudotime tended to have smaller hemisphere volume ($R = -0.18$, $p = 0.00012$) (Fig. 6d), and hippocampal atrophy ($R = -0.31$, $p = 5.8e-12$) (Fig. 6e). The association of protein pseudotime and hippocampal atrophy persists even after adjusting for the hemisphere volume ($R = -0.26$, $p = 1.3e-8$) (Fig. S11). Conversely, the association of the hippocampus volume with RNA pseudotime was largely attenuated with accounting for the hemisphere volume ($R = -0.10$, $p = 0.03$) (Fig. S11). This indicates that the protein pseudotime is a representation of not only regional molecular changes, but also brain structural changes, especially in the hippocampus.

Because sex has been shown to affect AD pathology[15] and gene expression[16] differently in older adults, we also examined the impact of sex on pseudotime and detected no or subtle sex differences in pseudotime. It is interesting because we have detected a strong sex effect on estimated protein abundance, indicating that our trajectory learning approach has removed the sex bias in the molecular signature and effectively aligned pseudotime with the rate of cognitive decline.

**Early proteomic events in ADRD progression**. The confluence of predicted protein abundance and pseudotime analysis provides a way to estimate the progression of proteomic changes from cross-sectional data. To identify early molecular events in ADRD progression, we first examined the pseudo-temporal progression of cognitive impairment and AD pathologies. The model predicts that amyloid deposition occurs first, then followed by PHFtau deposition, and finally, cognitive impairment (Fig. 7), which aligns with previous findings from positron emission tomography-imaging and cerebrospinal fluid measurements[17]. Then, we applied the same approach to the protein modules (Supplementary Data 7) and divided protein modules into early (amyloid), mid (PHFtau), or late modules (cognitive impairment) that altered 25% or more of the overall variation at the corresponding stages (Fig. 7). Modules that showed non-monotonic changes are termed "unspecified" modules. Notably, glial cells, including astrocytes, microglia, and oligodendrocytes, are primarily implicated in early modules, whereas excitatory and inhibitory neurons are involved in both early, mid, and late modules. The functions of early modules are related to mitochondria, synapses, protein folding, and cell defense, whereas mid and late modules are related to translation, RNA-binding, and transporting vesicles (Fig. 7 and Supplementary Data 8). The early modules are particularly interesting as they showed changes prior to cognitive decline. Interestingly, neuronal modules decrease their activities at the early- and mid-stages, whereas glial modules increase. It is worth noting that these modules cannot be identified as the top key modules for ADRD progression via the standard linear correlation approach because it assumes linear relationships between modules and cognitive impairment, thereby prioritizing mid and late modules (Fig. S12). The eigenvalues for late modules calculated from raw RNA expression data also showed the strongest associations than other modules, but weaker associations than those of estimated proteins.

## Discussion

We developed a proteome prediction model in order to overcome a major limitation of transcriptional studies that are intended to drive protein drug target discovery. Our approach allowed us to prioritize differentially expressed transcripts with concordant

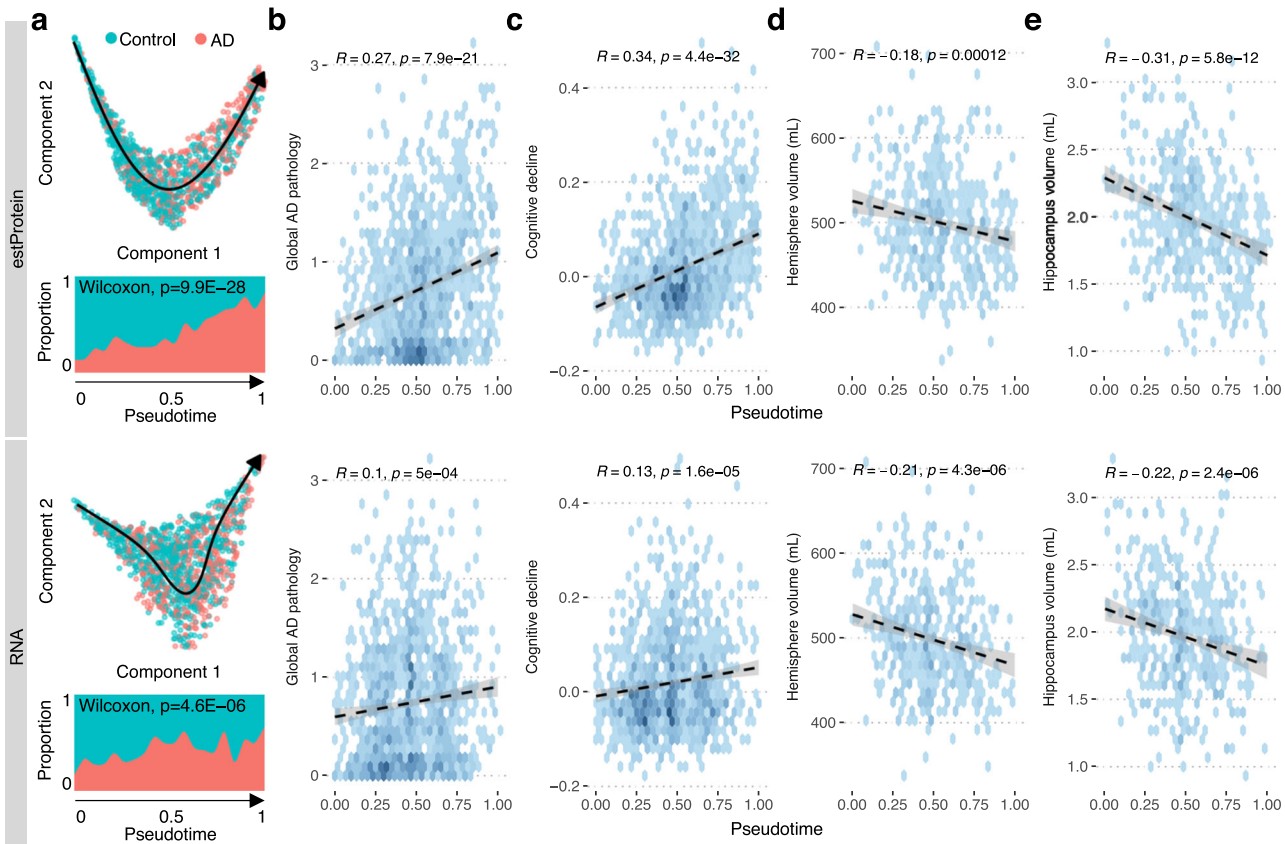

**Fig. 6 Proteome-based pseudotime captures the progression of dementia. a** The trajectory of brain proteome and transcriptome in older adults. Each individual was mapped to 2-dimensional space using the spectral embedding algorithm given the top 40 PCs of the estimated proteome. Then, we used SCORPIUS to estimate the trajectory of the protein system across individuals and pseudotime for each individual. The stream plot on the bottom shows that the proportion of AD people is increased as the pseudotime proceeds. The density plots represent relationships of **b** global AD pathology, **c** cognitive decline, **d** hemisphere volume, and **e** hippocampus volume with pseudotime based on estimated proteome and actual transcriptome. Pearson's correlation and *p*-value are indicated in each plot. The shaded areas represent the 95% confidence level interval for predictions from a linear model.

protein changes and also identify ADRD-associated proteins with negligible changes at the transcript level. This accuracy is likely mediated by mRNAs informing the abundance of protein machinery involved in translational processes (Fig. 3b). For instance, the elevation of immune-related proteins and the reduction of mitochondrial proteins were more evident at the protein level (Fig. 5b). We found that known AD risk genes such as APP and PICALM were strongly differentially abundant in their estimated protein levels, but not at their mRNA levels (Fig. 5c). Thus our integrative method complements standard RNA-seq analysis and bridges the gap between disease genetics and post-mortem omics.

Due to the complexity of gene regulation, the discrepancy between mRNA and protein levels has been a standing question since paired genome-wide measurements became available[18]. Because this predictive model brings the two measurements significantly closer together, we are able to better understand the nature of their relationship by examining the components of the model. There are two types of mRNA-vs.-protein relationships: (i) across genes within a sample (ii) across samples for each gene. For example, the RNA-to-protein ratio was proposed to estimate the former relationship from RNA expression[19]. Our method modeled the latter relationship, which is usually difficult to investigate as it requires a large number of samples. Therefore, our model was developed using the largest human brain data, with paired mRNA and protein measurements. Although most proteins showed a positive correlation with their mRNA levels,

the overall correlation was quite low. Notably, this is not specific to a particular proteomics technology themselves, or limited to a specific cohort, as we observed consistent results with three proteomic technologies and three independent cohorts. Therefore, the discrepancy is likely due to the various technical noises, post-transcriptional regulation, and translational activity for individual genes.

Although the disease signals showed a striking agreement between the estimated and actual proteomes (Fig. 4a), the overall correlation of protein levels is relatively low. To explore the theoretical limit of the accuracy of protein prediction, we compared two protein datasets from the same brain region in the same individuals ($N = 384$) measured by TMT and SRM. Even though the majority of proteins showed a positive correlation, an average correlation between protein abundance quantified by SRM and TMT methods is as low as 0.20 (Fig. S13 and Supplementary Data 9). This low correlation value is likely due to the various technical noises introduced in the multi-step sample preparation and quantification procedures applied at scale to postmortem brain tissues. However, this low correlation does not limit the ability to investigate disease effects with our data as the correlation of fold change between AD and controls is quite high ($R = 0.88$, Fig. S14 and Supplementary Data 9). This seemingly contradictory observation can be explained if we assume that protein abundance is generated from the following model,

$$p = bx + N(0, \sigma^2),$$

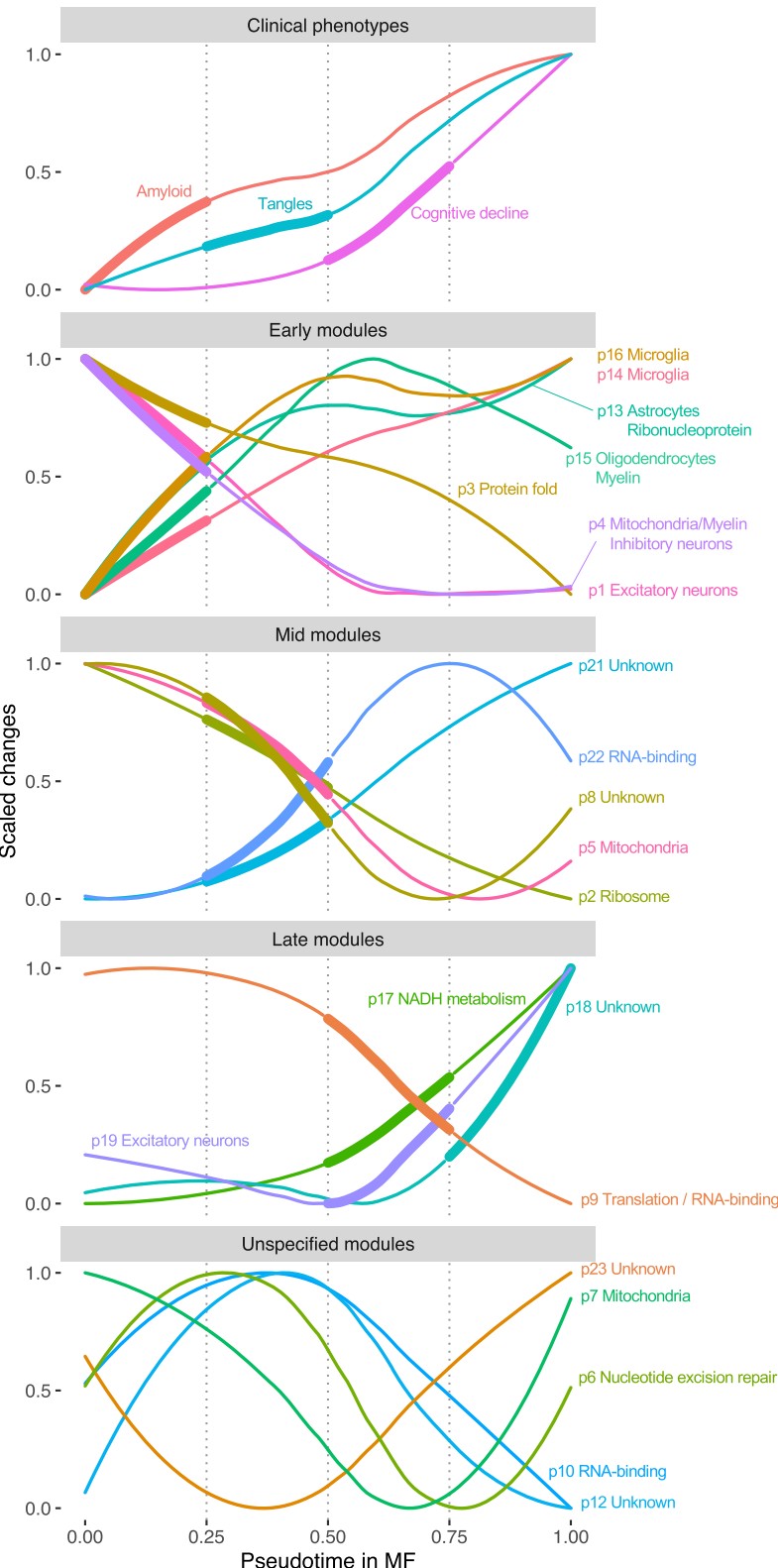

**Fig. 7 Pseudo-temporal ordering of cognitive dysfunction, brain pathologies, and protein modules.** Protein modules and their expression levels were computed using SpeakEasy. Pseudo-temporal changes of clinical phenotypes and module expression were smoothed using LOESS (local polynomial regression) curve fitting. Protein modules were grouped into early, mid, and late based on the period when they altered 25% of the overall variation, which was indicated as a thick line. Protein module annotations were based on enrichment of gene ontology and brain cell-specific genes for proteomic modules.

where $p$ represents measured protein abundance, $x$ is a phenotype of the sample, $b$ is an effect of the phenotype $x$ on the protein abundance $p$, and the last term is a noise following the normal distribution with zero mean and variance $\sigma^2$. The noise level greatly affects the correlation of protein abundance between two repeated measurements. However, even if the noise level is high, $b$ can be estimated with a decent accuracy because the measurement noise is independent of the phenotype. These empirical and theoretical backgrounds indicate that our predicted protein data from RNA-seq holds a reasonable accuracy to investigate protein systems in ADRD.

Although our model performs well with brain data set from an external cohort, the result may not guarantee that the model can work for any transcriptome data. The model takes PCs of transcriptome features or highly variable genes to look for low-dimensional latent space. However, PC structure or variability of gene expression could be very different among unrelated tissues. Thus, the scope of the model is likely limited to older brains, and re-training of the model in each tissue will be necessary to learn the tissue-specific transcriptome-proteome relationship. Down-sampling suggested that 100 samples are needed to obtain a 40% increase in detected proteins (Fig. S4). Thus, despite the necessity of acquiring some amount of paired RNA-seq and protein samples, the predictive power of the model is sufficiently strong to plan a cost-effective multi-omics study, wherein most protein data is estimated from RNA-seq.

In addition to its predictive accuracy, training the clei2block model is significantly faster than the Elastic-net and the CatBoost. To train the Elastic-net model with 5000 variables in the encoder input, it took around 9 s per protein using a single core of Intel(R) Xeon(R) CPU @ 2.30 GHz, and thus 20 h for all 8391 proteins. In addition to this, the 10-fold-cross validation experiment requires 10 times more of this, which is 200 h of computing time in total. For the CatBoost model, it took around 435 s per protein (1000 h for all proteins) and thus will require more than 10,000 h of computation in total. On the other hand, the clei2block model only took 9 min to train and thus 1.5 h in total using the same CPU with an NVIDIA Tesla T4.

There are likely multiple reasons why we found more proteins associated with AD phenotypes than mRNAs. First, protein levels likely reflect cognitive decline more than transcript levels, and the estimated proteome successfully gained this phenotypical corollary. The actual data supported this hypothesis as we detected more differentially expressed genes at protein levels than mRNA levels in the training cohort. Second, the scGen component in the clei2block model potentially reduces the measurement noise in data. The reason for that is when the model is properly trained, it will not be able to learn noise that is independent between protein and mRNA measurements and between different genes. Thus, the predicted protein should have less noise. If this is the case, it may be one source of the stronger statistical significance we observe with AD phenotypes.

The increased relevance of the predicted proteome to cognitive decline facilitated pseudotime trajectory analysis that translates all of the adjusted mRNA effects into a global estimate of brain state. While a pseudotime trajectory is estimated without a priori reference to any phenotypes, it explained 11.2% of the rate of cognitive decline independently from demographics and RNA pseudotime, indicating that protein pseudotime likely captures a significant component of cognitive decline (Fig. 6c). As the pseudotime reflects collective changes among hundreds of proteins, our result indicates that the brain may cope with the deleterious effects of ADRD by coordinating global protein systems rather than individual proteins. Interestingly, the protein trajectory progression was also associated with atrophy of the hippocampus region, an early inflicted region in ADRD (Fig. 6e).

Thus, the predicted protein trajectory model may provide a unified representation of the molecular, structural, and cognitive progression of ADRD.

In summary, deep neural network-based prediction of protein abundance helps address a long-standing biological conundrum and also practically provides the basis for improved systems biology analysis of ADRD data. Specifically, we show how it facilitates common coexpression and pseudo-temporal approaches, thereby producing more accurate estimates of the protein systems with early causal effects on ADRD progression.

## Methods

**Ethical statement**. The ROS and MAP studies were each approved by an Institutional Review Board (IRB) of Rush University Medical Center. Both studies enroll older persons without known dementia. All participants agree to an annual detailed clinical evaluation and organ donation at the time of death. Prior to enrollment, each participant signed informed consent and an Anatomical Gift Act for (AGA) donation of the brain, spinal cord, nerve, and muscle to the investigators for research purposes. The AGA is recognized in all 50 states and the District of Columbia. It is an advanced directive that foregoes the need to obtain consent for autopsy from a next of kin at the time of death. Participants also sign a repository consent to allow their data and biospecimens to be shared in accordance with procedures established by the relevant IRB.

**Population characteristics**. All human subjects are participants in one of two prospective studies of aging (ROS and MAP). ROS/MAP is a community-based cohort study. As community-based cohorts, ROS/MAP has much less referral bias, which can introduce significant sociodemographic, clinical, and genetic differences into studies of patients. At enrollment, mean education was 16.2 years, 67.8% were female, 98.4% were non-Latino white, and 23.2% had one or more APOE 4 alleles. While all were without known dementia at study enrollment, 7.8% met a research diagnosis for dementia at their baseline evaluation. These are community-based observational studies. They are not our patients. Information on other diagnoses and treatments is limited. The mean age at death was 89.6 years and 35.9% were diagnosed with Alzheimer's disease at the time of death.

**Recruitment**. The ROS study is comprised of older catholic priests, nuns, and monks throughout the USA. The MAP study recruits older laypersons from the greater Chicago area. Participants are not compensated for their participation. All visits and data collection, other than optional biennial MRI, are performed as home visits to ensure convenience for the participant and data close to death.

**Cognitive and clinical evaluations**. AD diagnosis was based on criteria of the joint working group of the National Institute of Neurological and Communicative Disorders and Stroke and the Alzheimer's Disease and Related Disorders Association (NINCDS/ADRDA) as previously reported[20]. Uniform structured cognitive and clinical evaluations are administered each year by examiners blinded to data from prior years. Briefly, the cognitive battery contains a total of 21 cognitive performance tests, of which 19 are used to construct a global composite measure of cognitive function[21]. The longitudinal rate of decline was computed for each participant using linear mixed models with adjustment for the effects of age, sex, and education, which estimate person-specific residual slopes[21].

**Tau and beta-amyloid measurement**. Tissue was dissected from eight brain regions to quantify the burden of parenchymal deposition of beta-amyloid and the density of abnormally phosphorylated paired helical filament tau (PHFtau)-positive neurofibrillary tangles. Tissue sections (20 μm) were stained with antibodies against beta-amyloid protein and PHFtau protein, and quantified using image analysis and stereology, as previously described[21,22].

**RNA-seq gene expression**. ROSMAP RNA-seq data of DLPFC, PCG, and AC regions were downloaded from the AMP-AD platform (syn3388564)[4]. To increase the number of samples that have both TMT proteomics and RNA-seq data, we sequenced additional 63 TMT cases without RNA-seq data at the Rush Alzheimer's Disease Center. RNA was extracted using Chemagic RNA tissue kit (Perkin Elmer, CMG-1212). RNA was concentrated (Zymo, R1080) and RQN (RIN score) was calculated using Fragment Analyzer (Agilent, DNF-471). RNA concentration was determined using Qubit broad range RNA assay (Invitrogen, Q10211) according to the manufacturer's instructions. Totally, 500 ng total RNA was used for RNA-Seq library generation and rRNA was depleted with RiboGold (Illumina, 20020599). A Zephyr G3 NGS workstation (Perkin Elmer) was utilized to generate TruSeq stranded sequencing libraries (Illumina, 20020599) with custom unique dual indexes according to the manufacturer's instructions with the following modifications. RNA was fragmented for 4 minutes at 85 °C. First-strand synthesis was extended to 50 min. Size selection post adapter ligation was modified to select for larger fragments. Library size and concentrations were determined using an NGS

fragment assay (Agilent, DNF-473) and Qubit ds DNA assay (Invitrogen) respectively, according to the manufacturer's instructions. The modified protocol yielded libraries with an average insert size of around 330–370 bp. Libraries were normalized for molarity and sequenced on a NovaSeq 6000 (Illumina) at 40–50 million reads, 2× 150 bp paired-end. RNA-Seq data processing was implemented using three parallel pipelines, an RNA-seq QC pipeline, a gene/transcripts quantification pipeline, a 3′ UTR quantification pipeline. In the QC pipeline, paired-end RNA-Seq data were first aligned by STAR v2.6[23] to a human reference genome. The primary assembly of reference genome fasta file and transcriptome annotation came from Gencode (Release 27 GRCh38). Picard tools were applied to the aligned bam files to assess the quality of RNA-Seq data. In the quantification pipeline, transcript raw counts were calculated by Kallisto (v0.46)[24]. To quantify pre-mRNA abundance, the transcriptome reference was customized. Transcript counts were aggregated at the gene level to obtain gene counts separately in mRNAs and pre-mRNAs. Samples were excluded if the total reads mapped were less than 5 million. A total of 17,294 mRNAs and 17,804 pre-mRNAs were expressed in >50% of the sample with at least 10 counts in each sample. These genes were included for the downstream normalization process. First, the CQN (conditional quantile normalization) was applied to adjust a sequence bias from GC content and gene length[25]. Next, the adjusted gene counts matrix was converted to log2-CPM (counts per million) followed by quantile normalization using the voom function implemented in the limma R package[26]. Finally, an LR model was applied to remove major technical confounding factors, including post-mortem interval, sequencing batch, RQN (RNA quality number), total spliced reads reported by STAR aligner, and metrics reported by Picard and Kallisto. To quantitate the relative proportion of 3′ UTR isoforms in each gene, we used QAPA[27]. A normalized 3′ UTR length was computed as the percentage of isoform-weighted 3′ UTR length[28]. We filtered out genes with missing UTR length in more than 100 samples or with identical UTR length in more than 95% of samples, resulting in 3204 genes. UTR length was log2-transformed, and then technical confounding factors were removed as described above.

**Mass spectrometry-based proteomics using isobaric TMT.** We utilized ROS-MAP TMT proteomics data from frozen tissue of the DLPFC generated by other researchers and their methods are already published in multiple papers in detail[4,29]. Briefly, digested protein samples were labeled with isobaric TMT and fractionated by high pH liquid chromatography (LC). Fractions were then analyzed by LC–MS. The resulting MS spectra were searched against the Uniprot human protein database and quantified. The effect of the experimental batch and PMI on quantified protein abundance were regressed out via LR. Proteins with missing values in more than 50% of the 384 subjects that overlapped with RNA-seq samples were excluded. A total of 8,391 proteins in 384 persons passed the final QC.

**Targeted selective reaction monitoring (SRM) proteomics.** SRM data were downloaded from the AMP-AD knowledge portal (10.7303/syn10468856)[30]. The SRM proteomics was performed using frozen tissue from the DLPFC region for proteins suggested by the consortium members of AMP-AD. We used 121 genes corresponding to 171 peptides that were measured both in the TMT and RNA-seq measurements. The samples were prepared for LC-SRM analysis using the standard protocol[30]. The abundance of endogenous peptides was quantified as a ratio to spiked-in synthetic peptides containing stable heavy isotopes. The "light/heavy" ratios were log2 transformed and shifted such that the median log2-ratio was zero. Finally, an LR model was applied to remove technical confounding factors including PMI and experimental batch.

**Deep-neural protein translation model (clei2block).** To estimate the tissue state that can inform the global proteome profile, we modified the scGen framework[11] that is designed to find the latent state representation of the transcriptome profile. Specifically, each transcriptome profile is encoded into low dimensional probabilistic distributions and then the decoder network takes an encoded vector sampled from the distributions to create the corresponding proteome abundance. Then, this decoded proteome was merged with the transcript features of each protein via an LR layer to generate a predicted proteome profile. The overall model architecture will be found in Fig. S15. Since translational efficiency is affected by variations in mRNA structure, we used mRNA abundance, pre-mRNA abundance, and 3′ UTR length as input transcriptional profiles. To reduce the dimension of the transcriptional profiles, we used the 5000 most variable transcriptional features or top 100 PCs. For this feature selection process, we used RNA-seq data from samples without TMT measurements to prevent information leakage from training data. We trained 12 models with a different combination of inputs and we then averaged all model predictions with equal weight to make a consensus prediction (Figs. S1 and S2). This ensemble process greatly improved model performance (Fig. 2b, c).

For training the clei2block model, we split the samples into ten folds by balancing cognitive status and brain pathologies. Then, for each fold, we removed the fold from the data and used the remaining data to train the clei2block model. Within the training samples, 10% of them were used as validation data to monitor learning progress and select the best model during the iteration. The trained model was applied to the holdout testing data so that we obtained predicted protein data

for the holdout samples. We repeated this process for each of ten holds, resulting in obtaining ten models trained with different training samples and holdout testing data. To calculate overall prediction accuracy, predictions for the holdout samples were concatenated after scaling within each hold and then contrasted with actual proteome abundance with Pearson's correlation (Fig. S3). This procedure ensures no information leakage from the holdout samples.

Model parameters were optimized using Adam optimizer with a learning rate of 0.001 and 0.01 for the scGen module and the LR module, respectively. The number of maximum training epochs was set to 10,000 with an early-stopping of 30 based on validation loss. The clei2block model is implemented in Pytorch and trained using an NVIDIA Tesla T4 GPU with the public docker image for Pytorch (v1.2) on Google Cloud Platform.

**Elastic net model.** The elastic-net-based protein prediction model was built based on the same data used for training the clei2block model for each protein separately. First, two hyperparameters, alpha, and lambda were tuned via fivefold cross-validation. Then with the hyperparameters showing the best RMSE (root-mean-squared error), we trained the elastic-net model using the whole training data. Lastly, the trained elastic-net model was applied to the hold-out test samples to generate predicted protein abundance. The elastic-net model was generated using the glmnet R library (v4.1-2).

**CatBoost model.** The CatBoost model was built using the catboost R library (v0.21) with the default setting. We used 10% of the training samples as validation data. We tuned the number of trees based on the RMSE of validation data with the maximum number of trees of 10,000 and early stopping of 50 iterations. Because the computation time for the CatBoost was significantly longer than elastic net or neural networks, we did not optimize other hyperparameters. Due to this limitation, the performance of CatBoost might be underestimated.

**Model validation using data from MSBB cohort.** To validate model prediction with data from an external cohort, we downloaded RNA-seq data (syn8612191) paired with label-free proteomics data (syn8495241) measured in the frontal pole (Brodmann area 10) from 196 participants of MSBB cohort[31]. The demographic of this cohort is similar to that of ROSMAP. For instance, the majority of participants are white, and their average age of death is at least 83, whereas their age over 90 has been censored to 90. About 63% of those are female and 51% are diagnosed as AD based on the CERAD criterion. Fastq files were processed using the same pipeline for the ROSMAP cohort to obtain normalized mRNA, pre-mRNA, and 3′ UTR length values. To correct batch and technical effects, we removed the sequencing batch, RIN, PMI from these transcriptional matrices. Label-free quantification of 3415 proteins for the same individuals was normalized so as to remove the effect from the experimental batch and PMI using the script in the Synapse repository (syn8495241). We applied each submodel trained with ROSMAP data to MSBB data and then averaged model prediction. Then we compared the predicted protein levels with the actual ones for 3222 proteins measured both in ROSMAP and MSBB data.

**Coexpression analysis.** To statistically identify protein modules, we used a consensus clustering approach SpeakEasy[32]. We computed the normalized protein level by subtracting the mean level for that variable across all individuals and dividing it by the standard deviation. Then, we summarized the composite metric of each module in each individual by computing the mean of the normalized levels across all variables in that module. To annotate protein modules, we performed enrichment analysis for GO terms and brain cell-specific gene sets using Fisher's exact test. The cell-specific genes were obtained from the single-nucleus RNA-seq of the DLPFC region from the ROSMAP cohort[16]. Specifically, reads from each cell type were summed to create pseudo-bulk RNA-seq data, and then for each cell type genes expressed more than twofold compared to other cell types were defined as cell-specific. To examine whether protein modules are preserved in transcriptome or estimated proteome, we ran modulePreservation function implemented in WGCNA R package[33]. The network type was set to signed and other parameters were set to default.

**Key predictors.** To evaluate the importance of input features to the prediction, we calculated the SHAP score using GradientExplainer[34]. The GradientExplainer method estimates the contribution of each input based on the difference in the gradient of the input from the background input distribution[35]. To compute the SHAP score, we used the model that takes PCs of mRNA features and mRNA, pre-mRNA, and UTR length because the independence of input features is desirable for an accurate estimate of variable contribution. The SHAP score was computed for testing data of 100 random proteins. To obtain robust estimates of SHAP score, we averaged SHAP scores from models built from different subsets of data for each submodel. We ranked PCs based on the average absolute SHAP score. To understand the key biological system for the prediction, we conducted GO enrichment analysis for genes whose loading of the PCs exceeded 95 percentile of all genes. The enrichment analysis was conducted using Fisher's exact test with GO terms in MSigDB v6.1[36].

**Protein interaction networks, transcription factors, and RNA binding proteins**. We downloaded the genomic locations of the binding sites of 171 RBPs from POSTAR2[37] as of October 2018 and 833 TFs from GTRD[38] as of October 2018. We then counted the number of RBPs, and TFs bound to promoters and exons. A promoter region of each gene was defined as the region from 2000 bp upstream of the transcriptional start site (TSS) to 1000 bp downstream of the TSS. Protein-protein interaction networks were downloaded from eXpression2Kinases Web[39], which consists of 209,459 interactions across 15,452 proteins. The degrees of proteins in the networks were computed using the igraph R package (v1.2.6).

**Transcriptome/proteome-wide association study**. An LR model was used to test the associations of transcripts or proteins with a continuous or categorical outcome. Age at death, sex, and years of education were included in the model as covariates. Bonferroni correction was employed to reduce the false positive due to the multiple testing.

**Molecular systems contributing to disease association**. To estimate the SHAP score, which is the feature contribution to predicted protein abundance, we applied GradientExplainer to the model with mRNA-PCs and all protein-specific inputs. The SHAP score satisfies the properties of local accuracy where the sum of feature contribution matches the original model output. With this property, we can evaluate how each feature affects the result of differential expression by subtracting feature contribution from the original prediction. We calculated SHAP scores for AD GWAS genes for the 808 samples that were not used for the model training. For each gene, we computed the variance of protein abundance explained by AD diagnosis, cognitive decline, and global AD pathology using the original estimated abundance and ones with the removal of the feature contribution for each mRNA-PC. We determined influential mRNA-PCs based on the difference in variance explained between them. To understand the key biological systems for the gene-disease association, GO enrichment analysis for each influential PC was conducted as described adobe.

**Trajectory analysis**. To infer the brain omics trajectory, we first mapped each individual into two-dimensional space based on the similarity of omics profile. To do this, we used the spectral embedding method implemented in the scikit-learn python library. Briefly, the k-nearest neighbor graph ($k = 10$) among individuals was constructed based on the Euclidean distance with the top 40 PCs of their omics profiles. Then, the structure of the graph representing individual-to-individual relationships was embedded into low-dimensional vectors by computing eigenvectors of its graph Laplacian. With the two-dimensional representation of the graph, we inferred the trajectory using SCORPIUS[14] which is one of the best performers in the recent method comparison[8]. Briefly, SCORPIUS partitioned samples into three clusters and optimized the shortest as well as the smooth path that goes through the center of clusters. Samples were projected onto the given path and pseudotime was assigned to each sample. The pseudotime estimates were highly robust with a range of k (5 to 50) and the number of PCs (30–60) as the average Spearman's correlation between estimates was 0.99. Based on the pseudotime, smoothed pseudo-temporal changes of clinical phenotypes and protein module expression were estimated using LOESS (local polynomial regression) curves, fitting with a degree of two. The proportion of variance in cognitive decline explained by pseudotimes and demographics was computed using relaimpo R package.

**Brain imaging**. At approximately 30 days postmortem, cerebral hemispheres were imaged in a 3-Tesla MRI scanner using previously described techniques[40,41]. Images were warped to a previously developed postmortem cerebral hemisphere template, on which we had manually drawn a mask encompassing the hippocampus. We back-transformed this mask onto images in their original space by applying the inverse of the individual-to-template transform to the mask image, as previously described[42,43]. After eliminating non-tissue-containing voxels, we extracted the resultant volume of each back-transformed mask, yielding a measure of hippocampal volume. The total hemisphere volume was calculated based on the number of tissue-containing voxels.

**Reporting summary**. Further information on research design is available in the Nature Research Reporting Summary linked to this article.

## Data availability

The RNA-seq and protein data used in these analyses, the predicted proteome data, and the estimated pseudotimes are distributed under the controlled data restrictions with a requirement of the Data Use Agreement. The data can be requested at the RADC Resource Sharing Hub at www.radc.rush.edu or the AD Knowledge Portal (https://adknowledgeportal.org) with the following accessions: ROSMAP RNA-seq (syn3388564, 10.7303/syn3388564), ROSMAP TMT data (syn17015098, 10.7303/syn17015098), ROSMAP SRM data (syn10468856, 10.7303/syn10468856), MSBB RNA-seq (syn8612191, https://www.synapse.org/#!Synapse:syn8612191), and MSBB label-free proteomics data (syn8495241, https://www.synapse.org/#!Synapse:syn8495241). The AD Knowledge Portal is a platform for accessing data, analyses, and tools generated by the Accelerating Medicines Partnership (AMP-AD) Target Discovery Program and other National Institute on Aging (NIA)-supported programs to enable open-science practices and accelerate translational learning. The data, analyses, and tools are shared early in the research cycle without a publication embargo on a secondary use. Data are available for general research use according to the following requirements for data access and data attribution (https://adknowledgeportal.synapse.org/DataAccess/Instructions). RNA-binding regions are available at POSTAR2 (http://postar.ncrnalab.org). Transcription factor binding regions are available at GTRD (https://gtrd.biouml.org). Protein-protein interaction networks are available at eXpression2Kinases Web (https://maayanlab.cloud/X2K/). Gene ontology is available at MSigDB (https://www.gsea-msigdb.org/gsea/msigdb/).

## Code availability

The deep-neural protein translation model is available at https://github.com/stasaki/clei2block[44] and https://doi.org/10.7303/syn23624037. The code for trajectory analysis is available at https://github.com/stasaki/SCORPIUS[45].

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

## Acknowledgements
We thank the study participants in ROSMAP and the staff of the Rush Alzheimer's Disease Center (RADC) and with support and data generation from NIH/NIA R01AG057911 (CG), R01AG061798 (CG), P30AG10161 (DAB), R01AG017917 (DAB), R01AG015819 (DAB), R01AG034374, K25AG061254 (RJD), and R01AG042210. The results published here are in whole or in part based on data obtained from the AD Knowledge Portal (https://adknowledgeportal.synapse.org). Study data were provided by the Rush Alzheimer's Disease Center, Rush University Medical Center, Chicago. Data collection was supported through funding by NIA grants P30AG10161 (ROS), R01AG15819 (ROSMAP; genomics and RNAseq), R01AG17917 (MAP), R01AG30146, R01AG36836 (RNAseq), U01AG32984 (genomic and whole exome sequencing), U01AG46152, U01AG61356 (ROSMAP AMP-AD, targeted proteomics), U01AG46161 (TMT proteomics), U01AG61356 (whole genome sequencing, targeted proteomics, ROSMAP AMP-AD), the Illinois Department of Public Health (ROSMAP), and the Translational Genomics Research Institute (genomic). Additional phenotypic data can be requested at www.radc.rush.edu. Study data were provided through the Accelerating Medicine Partnership for AD (U01AG046161 and U01AG061357) based on samples provided by the Rush University Medical Center, Chicago. Data collection was supported through funding by NIA grants P30AG10161, R01AG15819, R01AG17917, R01AG30146, R01AG36836, U01AG32984, U01AG46152, the Illinois Department of Public Health, and the Translational Genomics Research Institute. The results published here are in whole or in part based on data obtained from the AD Knowledge Portal (https://adknowledgeportal.synapse.org/). These data were provided by Dr. Levey from Emory University based on postmortem brain tissue collected through the Mount Sinai VA Medical Center Brain Bank provided by Dr. Eric Schadt from Mount Sinai School of Medicine.

## Author contributions
S.T. conceptualized and designed the study; D.R.A., L.J., and Y.W. contributed to generating RNA-seq data; R.J.D. processed MRI data. S.T. and J.X. processed RNA-seq data; V.A.P. generated and processed SRM data; S.T. constructed the predictive models and conducted systems biology analysis; S.T., V.A.P., R.J.D., Y.W., and C.G. interpreted the result; D.B. and C.G. each secured funding and supervised resource allocation and data generation; S.T. wrote the first draft of the paper; S.T., D.B., Y.W., and C.G. made a major contribution to the paper revision; all authors edited, read, and approved the submitted version.

## Competing interests
The authors declare no competing interests.
