## [Peer Review File · Nature Communications]

Inferring protein expression changes from mRNA in Alzheimer's dementia using deep neural networksReviewers' Comments:

Reviewer #1:

Remarks to the Author:

Tasaki et al. developed a deep learning approach to predict quantitative proteomics data from the transcriptome using matched data from hundreds of samples from AMP-AD. They then apply their model to >1,000 RNA-seq profiles from AMP-AD and describe associations of proteins and protein co-expression modules with AD-related traits. I find the general strategy of predicting the proteome from the transcriptome to be very compelling and novel, and the machine learning model appears well constructed, using cutting edge methodologies. My most fundamental concern is that the prediction accuracy is very low for many of the genes included in the downstream analyses, suggesting that there is a high degree of noise. One solution to this would be to focus on predicted values for a smaller number of proteins that can more accurately be predicted. In addition, given the inherent noise in the predictions, it is troubling that there is no experimental validation to support any of the analyses, and I would like to see direct experimental validation for at least 1-2 proteins for which novel associations with AD are predicted. In addition, I would like to see several additional analyses to confirm the robustness of the authors' conclusions with respect to Alzheimer's disease, described below section by section.

1. Model specification and initial performance evaluations. As mentioned above, my most fundamental concern is that the threshold $FDR < 0.05$ is VERT minimal. According to Table S2 this corresponds to a correlation threshold of about just $r=0.05$. Of the ~5,000 proteins used in downstream analyses, >800 proteins had correlation < 10% for predicted vs. observed protein levels. Studying Table S2, it appears that the model performs substantially better for a fair number of proteins (e.g., $r > 0.3$ for 1,479 proteins). If there is some reason to justify the inclusion of weakly predicted proteins in the model, please state this clearly and back it up with additional analyses stratified by prediction accuracy. However, a better solution may be to focus on a smaller number of proteins that have accurate predictions. This only works if training set accuracy predicts test set accuracy. Along those lines, how well does training set performance predict holdout performance – i.e., Is there substantial falloff in performance from training to holdout sets? Can you select those proteins that had a training set accuracy above some *reasonable* threshold (e.g., $r > 0.3$) and move forward with those into the holdout set?

In addition, results on model parameters and performance (lines 108-121; Fig. 2a-c) are not described in sufficient detail. Specifically, results are summarized by the # of proteins that are significantly predicted at $FDR < 0.05$. I would like to see Fig. 2a-c replaced with violin plots (or similar) to show the range of correlation values. Also, it is stated that the average correlation for all proteins was 0.18. What correlation method is used? Is this in training, testing, or holdout? Finally, Table S2 does not appear to contain any data about the performance in the SRM and MSBB datasets. Please add this information.

It's great to see a strong correlation for predicted vs. observed differential expression statistics ($R = 0.88$, line 177). What accounts for the stronger correlation for these statistics than the person-to-person variability in protein abundance? Do the proteins with relatively weak performance in the model nonetheless contribute to these DEG statistics? If so, how do you interpret that result?

For the analyses in Fig. 2g, in addition to this information about the informative features in the model, it would be useful to know which kinds of proteins can or cannot be predicted. E.g., does the model do best with highly abundant, highly variable proteins? Are proteins with a longer half-life harder to predict than proteins with a shorter half life or vice versa? Etc.

A few additional model combinations would be interesting to see in order to better understand what is driving the performance: (i) Model output with all Encoder input. (ii) How does each of the Encoder input component independently performs for prediction? (iii) Similarly, how does all the LM input performs without any Encoder input?

2. Associations of predicted protein levels with rate of cognitive decline in AD (Fig. 3; Section beginning at Line 198). Overall, these results are compelling. However, I find it confusing that the

main phenotype analyzed is the rate of cognitive decline, as opposed to a metric such as the cognitive score, pathology score (Braak score), or simply AD case vs. control. The authors argue that the rate is the measure that one would want to modify with a therapy. Fair enough. However, since the rate of cognitive decline is not linear over disease progression -- as appears to be assumed by the model -- this measure is difficult to interpret in the absence of parallel analyses describing the main effects on cognition and pathology. Since those data are available for ROSMAP samples, I would like to see the authors repeat their analysis using additional contrasts (diagnosis, cognitive score, pathology stage) and compare these results to the effects they are reporting on the rate of cognitive decline. One example for how the cognitive decline metric can be confusing / hard to interpret is for PICALM. The authors report a negative association of PICALM estProtein with cognitive decline. They claim that opposite effects of mRNA vs. protein are consistent with the literature, apparently based on Ando et al. 2013. However, contrary to the authors' assertion, that paper showed that PICALM immunoreactivity is increased in LOAD vs. control microglia and neurons, concordant with increased mRNA levels. The only thing that Ando et al. show is decreased is full-length PICALM, since they show the protein is abnormally cleaved. It is hard to tell whether the results from the deep learning model are really in disagreement with the Ando et al. study, since Ando compared LOAD vs. controls, which may not be equivalent to the effect on the rate of cognitive decline. Please read the literature on PICALM more carefully to determine if the result is consistent with other findings.

3. Protein abundance for GWAS risk genes. It's interesting to see that these genes from AD GWAS are associated with cognitive decline more strongly at the protein level than at the RNA level. This result would be more convincing if the authors could show that the risk-associated SNPs at these loci are QTLs for the estProtein levels of these genes. If so, can it be demonstrated that the protein abundance has a mediating effect? Testing for QTL effects and mediating effects would more directly establish (or rule out) the connection from SNP to protein to cognition.

4. In the pseudotime analysis (Fig. 4), as with the analyses in Fig. 3, I would find it very helpful to see comparable analyses to test associations of pseudotime with cognitive score, pathology score and diagnosis.

5. Connecting modules to GWAS. Fig. 5c. Stratified LDSC is not optimal for Gene Set Enrichment Analysis, as searching for enrichments in large regions around each gene in the network lacks specificity. Please confirm that results are robust when using a more standard tool for GWAS gene set analyses, such as MAGMA. Fig. 5d. The PRS analysis was performed using an unusually small number of SNPs (just 159 SNPs). Typically, PRS analysis explains more variance and becomes more robust when thousands of SNPs are included in the prediction model. Can the authors demonstrate that this result is robust to the p-value threshold used in calculating the PRS?

6. The article lacks validation studies for any of the proteins. Since the correlations of estProtein with observed proteomics signal were often quite low, some direct experimental validation of key findings seems essential. Which proteins to validate is the authors' choice but might include hub genes from early, middle, and late modules described in Fig. 5. The examples from these modules mentioned in the Discussion (line 411) are very interesting but would be more convincing with validation.

Minor points

1. Figure 1 is really a graphical abstract. I recommend that this be moved to the supplement.
2. Much of the introduction is devoted to a summary of results that would be better suited for the first paragraph of the discussion or removed.
3. There are some missing details regarding the ensemble averaging procedure. How was averaging done? Were all of the underlying models assigned equal weight? Etc.

4. Please provide exact p-values on figure 2 (not $p < 2.2e-16$).

5. I find the annotation of "medial frontal" cortex to be vague. The dissections were for much more precise sub-regions. For instance, Yu et al. (2020) in JAMA Psychiatry describes the proteomics samples as being from DLPFC. Please state exactly which sub-regions were used and combined.

6. On Fig. 3c, I find the y-axis of this figure confusing relative to the text. APP has a negative value for estProtein, and PICALM has a positive value. But the text indicates that APP is associated with an increased rate of cognitive decline. I think I can piece together what is being shown, but it's not clearly stated. An additional 1-2 sentences in the figure legend are needed.

Seth Ament

Reviewer #2:

Remarks to the Author:

Remarks to authors:

The manuscript "Protein systems in Alzheimer's dementia, via deep neural network analysis of RNA expression" by Tasaki et al proposes a deep neural network model to predict protein abundance from RNA expression. The authors present a few examples that show promise. In particular, the authors show the application of their predictive model to identify proteomic signatures associated with ADRD but I have major concerns about this paper as well. I found it difficult to understand the experiments explained by authors (i.e. what are train/valid/test splits) and generally hard to follow. Most of the analyses are based on in-silico predicted data which I believe is hard to evaluate without real ground truth and more careful evaluations. Additionally, the importance of the different modules of the pipeline is not justified through an ablation study to show why those pieces improve the prediction. Finally, the metrics make it hard to understand how well or poorly their method is performing.

Major points:

1. While the prediction of protein abundance from RNA has been shown feasible with reasonable accuracy (Buccitelli and Selbach 2020), Fortelny et al found that this prediction also works when protein-to-mRNA ratios are calculated from randomly shuffled or identical mRNA levels (Fortelny et al. 2017). I would like to see further experiments from authors to demonstrate their predictions are meaningful and such results are not possible with random or identical mRNA levels.
2. The model design and existence of specific components are not well justified, for example, it is unclear that pre-mRNA and UTR length would increase perf a lot. Authors should indicate correlations between each of those and proteins. Or why do we need scGen there at all? Generally, It would be beneficial for the readers to see careful and extensive ablation studies to show the relevance of each part of the model.
3. 90-91: "To mitigate the discrepancy, we developed a predictive model of the proteome solely based on transcriptome data", I found this part negating the whole section if the correlation is already low and not explained in the RNA data how could such data predict the protein level explaining such variances?
4. It would be beneficial if authors add additional baseline or existing methods to compare their model against them to further benchmark the model.

Minor points:

1. Figure 3c, can you investigate the effect size here?
2. I looked into the GitHub repository trying to reproduce or generate some of the results of the paper but were unable to do so. It would necessary to have some tutorials for the users and allow them to reproduce the results of the paper (or at least some part of it)

3. Figure 5D and S12: authors need to report these associations for RNA as well.
4. Why have authors leveraged a trajectory inference method for scRNA-seq data for RNAbulk and proteomic data where a lot of assumptions might not hold?

References

Buccitelli, Christopher, and Matthias Selbach. 2020. "mRNAs, Proteins and the Emerging Principles of Gene Expression Control." *Nature Reviews. Genetics* 21 (10): 630–44.

Fortelny, Nikolaus, Christopher M. Overall, Paul Pavlidis, and Gabriela V. Cohen Freue. 2017. "Can We Predict Protein from mRNA Levels?" *Nature* 547 (7664): E19–20.

Reviewer #3:

Remarks to the Author:

This is a very important breakthrough study to address a major challenge that faced the field of AD research, especially with regard to the translational aspect of drug discovery. There has been a tremendous effort to generate RNA transcriptomics from post-mortem brain tissues and compare that to transgene mice models to better understand the trajectory of the AD pathology. However proteins carry out the actual work and are closer to the observed phenotype, but technical challenges have made large-scale proteomic studies lag behind their RNA-based -omics counterparts.

The authors use an innovative deep learning technique to develop a "protein-like" system analysis, based on a training set of concordant proteomic and transcriptomic data of a relative small subset of subjects in the ROSMAP database and apply this to the largest available database in the field. They show a limited overlap between the transcriptomics and "proteomic like" based prediction models and that the proteomic-like prediction model has a closer relationship with actual clinical phenotype such as hippocampal atrophy and cognitive decline.

The authors address very well the shortcomings of their approach in putative generalizability (although to their credit they use different datasets to validate their findings). Interestingly they identify some solutions to address these challenges with new datasets and patient populations.

The paper is well written and the experimental details provided in sufficient detail.

A few minor comments to add to the discussion:

1. It might be of interest to mention and contrast to other methods for predicting tissue-specific translation using large scale proteomics and transcriptomic data resources such as non-neural network approaches (e.g. Edfors 2016 <https://www.ncbi.nlm.nih.gov/pmc/articles/PMC5081484/>).
2. An interesting finding is the appearance of APP and PICALM in the protein-like prediction model but not in the RNAseq model, a fact the authors mention in their introduction. It would be of interest to discuss how these new proteomic markers would appear after the deep-learning approach. On the other hand, APOE the most robust genetic risk factor for AD has only a weak effect in the proteomic-like system analysis.
3. Post-translational modifications (phosphorylation, acetylation, glycation, truncation, etc) of many AD-related proteins drive the progression, which is obviously beyond the scope of this report.
4. Correlation or association is not per se causation; the biology is often more complex than a linear relation between expression levels and clinical phenotype (as has been painfully demonstrated in the amyloid modulating clinical trials). These approaches do not obviate the need for further biological research to really use this as a target selection tool.

Hugo Geerts, Certara

Reviewer #4:

Remarks to the Author:

Summary

Tasaki and colleagues implemented a deep neural net for the estimation of protein abundances from the mRNA abundance, pre-mRNA abundance, and 3'UTR length from the transcriptome data as input. The *in silico* translated data are further analyzed in three different ways, namely co-expression network analysis, association with cognitive decline, and pseudotime analysis that reflect the dementia progression. Despite seemingly interpretable results, it won't be a stretch to say that the validity of all reported findings hinges on successful prediction of protein abundance values. The authors make a case that the predicted protein abundance data improve the aforementioned systems-level inference compared to an analysis where mRNA abundance are regarded as proxy of protein abundance. From my point of view, however, these downstream implications are indirect metrics of success and do not validate the accuracy of proteome prediction. The detailed comments are as follows.

Major comments

1. The straightforward evaluation of the prediction performance should be based on the correlation between estimated relative protein abundance and experimentally measured protein abundance for individual proteins. In this sense, I think it's important to present the distribution of Pearson correlations between estimated proteome and targeted proteomics data in a histogram in the training data (TMT / N=384) as well as test data (against targeted proteomics data) in Figure 2. Through Figures 2a-2c, the authors use the false discovery rate for the hypothesis testing of H_0 : correlation = 0 and H_A : correlation \neq 0 as the evidence of improvement over the surrogate (mRNA), where the correlations are computed between predicted protein values and experimentally measured protein abundance, or between mRNA abundance and experimentally measured protein abundance. Using statistical significance measures of the correlations with actual measurements is equivalent to considering association as causality or prediction. Of course the authors have trained a prediction model in a way to increase the correlation with measured proteins, so the correlation increased. But how good or bad are the correlations?

2. In fact, I am sorry to say that I cannot lend much support to the prediction accuracy, which is briefly mentioned and undermines the credibility of the results. On page 4, line 110, it is stated that the average correlation between estimated protein abundance and TMT proteomics (cross-validated) for all proteins was 0.18. This means that you must have had a large number of proteins with nominal correlations close to zero or even negative. The first thing that comes to my mind is that you may need to filter some poorly predicted proteins from the estimated data.

3. Here is a more philosophical comment. If the proteome could be robustly predicted from the transcriptome data, that will be revolutionary. The irony here is that the authors describe the motivation of the work as the poor correlation between the transcriptome and the proteome, yet they infer the proteome from the transcriptome. Essentially, at the tissue level, the authors must be inferring that the amalgamation of all mRNA transcripts somehow yields good predictive information of quantitative levels of individual proteins, and this mechanism can be learned from a collection of a few hundreds of brain tissue samples.

I have a very hard time believing this. Deep neural networks are most successful in computer vision and image processing, where the data structure are values on grid coordinates on a lattice (e.g. pixels with a fixed number of neighbors), where the task is to identify certain patterns or types of objects. Even for these "simpler" tasks, convolutional networks or other types of deep learning methods often need to be trained on hundreds of

thousands, if not millions, of picture frames to attain reasonably error-free performance. Protein data are not on lattice grids and there is no regularity. Translation of each and every protein is governed by a different combinations of factors and cellular context, and therefore one can surmise that the protein abundance is likely a function of many biological inputs, not just the abundance of the template (mRNA) and the post-transcriptional and translation machinery in a specific tissue type. Do authors believe that they have sufficient training data to account for such complex mechanisms and claim that their deep neural net is not overtrained in their own data? If you can prove that you have very good prediction results (e.g. $R^2 = 0.9$ for thousands of proteins) with the presented data, this will be not an ordinary feat.

Minor comments

1. In line 27-29, I'm pretty sure we are way past the time when transcriptomic data are the best background to look for drug targets.
2. In Figure 1, it will be helpful to have more detailed information indicating what part of the samples were used as input to train the deep neural net (transcriptome + global proteome), and what other part of the samples with what data (e.g. transcriptome, validation by targeted proteomic data), with sample sizes indicated in appropriate places.
3. It is unclear why the authors chose regression with elastic net penalty as the only competing model. In predicting the abundance of protein A, was it possible for the mRNA features of the same gene to be omitted from the prediction model as a result of variable selection? Could you have chosen more sophisticated non-linear predictors like SVM with radial basis or tree-based gradient boosting / random forest?
4. Figure S3. What test statistics are you using? Can you add the info to the figure legend?
5. In lines 181-191, can you provide more detailed account to give insights as to how post-transcriptional regulation and translation machinery genes contributed to the prediction of abundance for individual proteins? Or do we just know that those genes had higher feature importance scores / causality scores?
6. It was observed in the paper that amyloid beta precursor (APP) gene itself was not differentially expressed in AD cases at the mRNA level, but the protein was (in TMT measured portion as well as in the estimated proteome). Were you able to explain what data features contributed the most to the elevated protein level of APP? Delineating this would be revolutionary, whether the increased APP level is driven by increased translation, or simply a defect in the neuronal clearance of amyloid beta.
7. A lot of details regarding proteomics experiments are lacking in the Methods section. I recommend the authors to include detailed supplementary information regarding all experimental parameters (both details of TMT-labeling-based global proteome analysis and the transitions / assay info of the SRM). The presence of supplementary information may allow the authors to reduce word counts by moving the description of clei2block to the Supp Info as well.

Reviewer #5:

Remarks to the Author:

The authors have developed a deep neural network model to predict protein abundance from RNA expression in aged brains. This would be of great interest to the scientific community, not only for Alzheimer's and neurodegenerative disease research but also for advancing the compatibility of transcriptomics and proteomics analyses. I believe the work should be published upon addressing my below concerns.

In focusing on the proteomics experiments, the already published TMT experiments appear to be well thought out and performed in a way consistent with how they are generally performed with the proteomics field. My concern is regarding the SRM experiments, as there is little mention to how they were performed in this work, and the references cited in the manuscript for the SRM experiments do not appear to involve SRM experiments at all. It is unclear if I am missing something or if the authors referenced the wrong papers.

Additionally, because of the potential interest from a broad audience, I believe a broader explanation of some items in the manuscript should be added. For example, the abbreviation "GWAS" should be explained.

Reviewer #6:

Remarks to the Author:

The paper "Protein systems in Alzheimer's dementia, via deep neural network analysis of RNA expression" presents a new approach to predict protein abundance from RNA expression, which is validated on multiple datasets. Three applications further demonstrate certain advantages of the predicted protein levels over mRNA signals. However, the current version of the paper does not provide enough explanation to the mechanism or enough benchmarking. The comments are listed below:

1. The clei2block extracts low dimensional latent factors from transcriptomes for model building, and the authors have tracked down the origin of the predictive accuracy from the protein specific input such as mRNA, pre-mRNA and UTR length and encoder input. However, the authors have not investigated the decoder output, which is used for linear model building as downstream input. Please provide more details on how the decoder output is different from the encoder input, how important is the scGen module in the clei2block pipeline?

2. The authors have made comparison with the Elastic-net model. Please compare clei2block with the model by only using linear regression module with the encoder input and protein specific input.

3. As the authors have mentioned that genes projecting to influential PCs were enriched for translational initiation and mRNA processing, such as ribosomal proteins, translation initiation factors, and splicing factors, which is rather general and not protein specific. Would these PCs introduce false positive in prediction? Please discuss this.

4. The authors mentioned that "We found 3826 proteins associated with cognitive decline (Bonferroni-corrected $p < 0.05$) (Figure 3a and Table S4), which is 15-fold more than those identified from actual proteome data of 384 samples and 2.4-fold more than those from transcriptome data of the same 1,192 samples". Please explain why the number of predicted proteins is much more than the actual proteome data in terms of association with cognitive decline. The more the better? Or is the actual proteome data unreliable?

5. The authors evaluate the accuracy of clei2block by comparing the number of mRNAs and estimated proteins positively correlated with actual protein abundance, which is not straightforward. Please

provide more details on prediction errors compared with the actual protein abundance.

6. The authors should address the scarcity of proteomics and highlight the advantages of transcriptome data over proteomics in the introduction section.

7. The authors mentioned that "Because training the Elastic-net model took a significantly long time with a large number of predictors", so how long does clei2block take to process the datasets in the manuscript. The author can address the computation efficiency of clei2block.

8. The clei2block can be used to identify ADRD-associated proteins with negligible changes at the transcript-level. For these proteins, which factors contribute mostly within the input? Please provide statistical analysis or references on how these proteins are influenced by the factors, such as ribosomal proteins, translation initiation factors, and splicing factors.

9. In Figure S15, where is the "Ensemble prediction" of scGen? Please provide more information.

10. In Figure 3a, the number of genes whose expression are associated with cognitive decline in AC dataset is much larger than the PCG dataset, while the number of genes whose estProteins are associated with cognitive decline in AC dataset is smaller than the PCG dataset. Is it due to batch effect? Please discuss this counterfactual phenomenon.

11. In the Fig 3c, it is suggested to place the gene symbol in the vertical axis and signed $-\log_{10}(P\text{-value})$ in the horizontal axis, making it more clear to readers.

12. In the Fig 3c, the gene ADAM10 is significantly associated with cognitive decline, while its estProtein is not. Please provide some possible explanations.

REVIEWER COMMENTS

Reviewer #1 (Expertise: Systems-approaches to studying neuro-psychiatric disorders, genomics):

Tasaki et al. developed a deep learning approach to predict quantitative proteomics data from the transcriptome using matched data from hundreds of samples from AMP-AD. They then apply their model to >1,000 RNA-seq profiles from AMP-AD and describe associations of proteins and protein co-expression modules with AD-related traits. I find the general strategy of predicting the proteome from the transcriptome to be very compelling and novel, and the machine learning model appears well constructed, using cutting edge methodologies. My most fundamental concern is that the prediction accuracy is very low for many of the genes included in the downstream analyses, suggesting that there is a high degree of noise. One solution to this would be to focus on predicted values for a smaller number of proteins that can more accurately be predicted. In addition, given the inherent noise in the predictions, it is troubling that there is no experimental validation to support any of the analyses, and I would like to see direct experimental validation for at least 1-2 proteins for which novel associations with AD are predicted. In addition, I would like to see several additional analyses to confirm the robustness of the authors' conclusions with respect to Alzheimer's disease, described below section by section.

1. Model specification and initial performance evaluations. As mentioned above, my most fundamental concern is that the threshold $FDR < 0.05$ is VERT minimal. According to Table S2 this corresponds to a correlation threshold of about just $r=0.05$. Of the ~5,000 proteins used in downstream analyses, >800 proteins had correlation < 10% for predicted vs. observed protein levels. Studying Table S2, it appears that the model performs substantially better for a fair number of proteins (e.g., $r > 0.3$ for 1,479 proteins). If there is some reason to justify the inclusion of weakly predicted proteins in the model, please state this clearly and back it up with additional analyses stratified by prediction accuracy. However, a better solution may be to focus on a smaller number of proteins that have accurate predictions. This only works if training set accuracy predicts test set accuracy.

Response: We appreciate the time that you have dedicated to this careful review and appreciate your encouragement. To explore the theoretical limit of the accuracy of protein prediction, we compared two protein datasets from the same brain region, in the same individuals (N=384), measured by tandem mass tag (TMT) system and selected reaction monitoring (SRM). Even though the majority of proteins showed a positive correlation, the average correlation between protein abundance quantified by SRM and TMT methods can be ~ 0.20 (**Figure R1**). However, the test set accuracy of our RNA-based prediction is actually close to the correlation value between TMT and SRM, which can be considered a practical ceiling on performance. Taking that context into account, the seemingly low correlation value is surprisingly high, given the various technical noises introduced in the multi-step sample preparation and quantification procedures through processing the large number of postmortem

brain tissues. Moreover, this level of correlation does not limit the ability to investigate disease effects with our data, which we show by a simulation study and actual data as below.

Simulation details proving phenotypic relevance of detected correlations:

We assume that protein abundance is generated from the following model,

$$p = bx + N(0, \sigma^2),$$

where p represents measured protein abundance, x is a phenotype of the sample, b is an effect of the phenotype x on the protein abundance p , and the last term is a noise following the normal distribution with zero mean and variance σ^2 . With this model, we generated two sets of synthetic protein data for 400 samples with the same parameters for each of 8,000 proteins. The phenotype values, x , for 400 samples and its effect on each protein, b , are sampled from the standard normal distribution. We then computed the correlation between two data sets and also examined the correlation of protein abundance with the phenotype. We varied the noise level, σ^2 , and examined how the noise level impacts the correlation of protein abundance between two independent measurements from the same individuals, and that of the estimated phenotype effect, b , across proteins. We repeated this experiment 10 times to assess the robustness of the result. As shown in **Figure R2**, the noise level greatly affects the correlation of protein abundance between the two measurements. However, the estimated b is highly robust between the measurements. This is because the measurement noise is independent of the phenotype and thus b can be estimated accurately even if the noise level is high. This is indeed the case for the real protein data of SRM and TMT proteomics (**Figure R3**). Even though the correlation of protein abundance between these two data is seemingly low, the correlation of fold change between AD and controls is quite high (R=0.88).

Furthermore, as the reviewer suggested, we stratified the proteins based on the test set accuracy and examined how the accuracy impacted the detection of disease signals (**Figure R4**). The result indicates that even for proteins with lower correlation, their disease effects are in line with the actual observation more than those of RNA. Based on these results collectively, our predicted protein data from RNA-seq holds reasonable accuracy to investigate protein systems in Alzheimer's dementia. We added these results in Figure 4, S12, and S13 in the revised manuscript.

Figure R1. Correlation of protein abundance measured by tandem mass tag (TMT) system and selected reaction monitoring (SRM). We quantified the abundance of 121 proteins in the mid-frontal region of 384 subjects using two independent technologies: TMT proteomics and SRM proteomics. The histogram represents the distribution of correlation between the protein abundance measured by TMT and SRM.

Figure R2. Relation of noise level and correlation between repeated protein measurements in protein abundance and phenotype effect.

Figure R3. Comparison of TMT and SRM datasets in log-fold change between AD and control.

Figure R4. Correlation between predicted and actual AD association stratified by prediction accuracy.

*Along those lines, how well does training set performance predict holdout performance – i.e., Is there substantial falloff in performance from training to holdout sets? Can you select those proteins that had a training set accuracy above some *reasonable* threshold (e.g., $r > 0.3$) and move forward with those into the holdout set?*

Response: As we showed in **Figure R1**, the performance limit with our protein data will be around R of 0.2, which is close to the value we obtained with the model. In addition, we validate our prediction model in the two separate data sets. Thus, a relatively low predictive accuracy

would not be due to excessive overfitting of the model to the training data but the nature of our protein data.

In addition, results on model parameters and performance (lines 108-121; Fig. 2a-c) are not described in sufficient detail. Specifically, results are summarized by the # of proteins that are significantly predicted at FDR < 0.05. I would like to see Fig. 2a-c replaced with violin plots (or similar) to show the range of correlation values. Also, it is stated that the average correlation for all proteins was 0.18. What correlation method is used? Is this in training, testing, or holdout? Finally, Table S2 does not appear to contain any data about the performance in the SRM and MSBB datasets. Please add this information.

Response: We added violin plots to show the distribution of correlations in Figure 2. Also, we revised Table S2 to include correlations for SRM and MSBB data as the reviewer suggested. We used Pearson's correlation and the performance metric is all from hold-out data, which are described in the method section.

It's great to see a strong correlation for predicted vs. observed differential expression statistics ($R = 0.88$, line 177). What accounts for the stronger correlation for these statistics than the person-to-person variability in protein abundance? Do the proteins with relatively weak performance in the model nonetheless contribute to these DEG statistics? If so, how do you interpret that result?

Response: As we showed above (**Figure R2**), when the noise is orthogonal to the phenotype affecting protein abundance, differential expression statistics are well correlated. Also, as shown in **Figure R4**, the prediction accuracy reflects the accuracy of DEG statistics, but the proteins with relatively weak performance still retain reasonable accuracy in DEG statistics estimates.

For the analyses in Fig. 2g, in addition to this information about the informative features in the model, it would be useful to know which kinds of proteins can or cannot be predicted. E.g., does the model do best with highly abundant, highly variable proteins? Are proteins with a longer half-life harder to predict than proteins with a shorter half life or vice versa? Etc.

Response: We agree it is important to understand the behavior of the prediction model. As the reviewer suggested, we conducted an extensive investigation in protein characteristics defining its predictability by RNA-seq (**Figure R5**). We found that the protein with higher mean intensity showed improvement in predictive performance by the model more than those with lower mean intensity. Conversely, the proteins with higher variance had less benefit from the model, as those proteins already have a higher correlation with their corresponding mRNA levels. We did not find a relationship between protein-half life and the model performance. A possible reason for the absence of association is that protein-half life data we found in the public domain does not cover all proteins we measured and is biased toward high-abundant and less variable

proteins (Figure R6). In addition to the metric the reviewer suggested, we explored various system-level molecular characteristics. We found that connectivity in protein co-abundance networks and protein-protein interaction networks, the number of transcription factors bound to the promoter region, and the number of RNA-binding proteins bound to mRNA adjusted with the length of mRNA were all positively correlated with the magnitude of performance improvement by the predictive model (Figure R5). Together these molecular characteristics explained 14% of the improvement in the predictive accuracy. Overall, our model performed well for the genes supposedly having relationships with many other proteins. The protein levels of such proteins might be regulated by or correlated with those of other proteins. Thus, it makes sense that accounting for expression levels of the regulators/correlates improves prediction accuracy, which our model essentially does with the encoder part. The result is included in the revised manuscript as Figure 3c.

Figure R5. Molecular and system characteristics defining protein predictability.

Figure R6. Half-life measurement is based on high-abundant and low-variable proteins.

A few additional model combinations would be interesting to see in order to better understand what is driving the performance: (i) Model output with all Encoder input. (ii) How does each of the Encoder input component independently performs for prediction? (iii) Similarly, how does all the LM input performs without any Encoder input?

Response: Thank you for your suggestion. We built the requested models and the result is shown in **Figure R7**. The model with all the encoder and LM input (All+All model) performed best among the non-ensemble models but worse than the ensemble model. Interestingly, adding the All+All model to the ensemble did not improve the performance and was even slightly worse than the ensemble without the All+All model. This indicates that merging outputs of models with a different combination of input is critical. Regarding each encoder input, mRNA and UTR models performed better than others, but again ensemble prediction achieved the best performance. Overall, dimensional reduction of input variables by PCA impaired the model performance. Lastly, the model with only LM input did not perform well, suggesting the importance of the encoder part. However, we want to note that when we looked at the contribution of each input variable instead of the encoder part as a whole, mRNA input in the LM part contributed to the prediction the most, and the clei2block models generally performed better than the encoder-only models. This indicates the LM part has a key role when we train the model with both encoder and LM jointly.

Figure R7. Performance of models with different combinations of inputs.

2. Associations of predicted protein levels with rate of cognitive decline in AD (Fig. 3; Section beginning at Line 198). Overall, these results are compelling. However, I find it confusing that the main phenotype analyzed is the rate of cognitive decline, as opposed to a metric such as the cognitive score, pathology score (Braak score), or simply AD case vs. control. The authors argue that the rate is the measure that one would want to modify with a therapy. Fair enough. However, since the rate of cognitive decline is not linear over disease progression -- as appears to be assumed by the model -- this measure is difficult to interpret in the absence of parallel analyses describing the main effects on cognition and pathology. Since those data are available for ROSMAP samples, I would like to see the authors repeat their analysis using additional contrasts (diagnosis, cognitive score, pathology stage) and compare these results to the effects they are reporting on the rate of cognitive decline.

Response: We added results for AD diagnosis and global AD pathology in Figure 5a. As cognitive score is well correlated with cognitive decline, we provided the result for cognitive

score as Figure S5. Overall, the trends of associations of these phenotypes are consistent with that of cognitive decline.

One example for how the cognitive decline metric can be confusing / hard to interpret is for PICALM. The authors report a negative association of PICALM estProtein with cognitive decline. They claim that opposite effects of mRNA vs. protein are consistent with the literature, apparently based on Ando et al. 2013. However, contrary to the authors' assertion, that paper showed that PICALM immunoreactivity is increased in LOAD vs. control microglia and neurons, concordant with increased mRNA levels. The only thing that Ando et al. show is decreased is full-length PICALM, since they show the protein is abnormally cleaved. It is hard to tell whether the results from the deep learning model are really in disagreement with the Ando et al. study, since Ando compared LOAD vs. controls, which may not be equivalent to the effect on the rate of cognitive decline. Please read the literature on PICALM more carefully to determine if the result is consistent with other findings.

Response: As the reviewer mentioned, Ando et al. 2013 reported PICALM levels for each isoform and it is not clear whether the total PICALM is decreased in AD. However, in their follow-up study in 2016, they reported that total PICALM levels are reduced in AD and other neurodegenerative diseases. Although they showed cell-type specificity in PICALM immunoreactivity in 2013, it is difficult to compare the cell-specific data with the TMT data as our proteomics data is based on bulk tissue. Thus, we believe that the most comparable experiments in Ando et al. 2016 are Fig4c or Fig5c as TMT proteomics data we used measured mainly soluble proteins. Since both Fig4c and Fig5c showed the reduction of PICALM in AD, our finding does not contradict the previous report. Interestingly, an inverse relation of mRNA and protein levels of PICALM to AD phenotype was also observed in the MSBB data. Specifically, mRNA levels of PICALM showed a negative correlation with cognition score (Clinical Dementia Rating Scale), but both actual and estimated protein levels showed a positive correlation. We think this is a good example where the clei2block model could reveal protein association to disease which is not suggested by mRNA levels.

Reference

Ando, K. *et al.* Level of PICALM, a key component of clathrin-mediated endocytosis, is correlated with levels of phosphotau and autophagy-related proteins and is associated with tau inclusions in AD, PSP and Pick disease. *Neurobiol. Dis.* **94**, 32–43 (2016).

3. Protein abundance for GWAS risk genes. It's interesting to see that these genes from AD GWAS are associated with cognitive decline more strongly at the protein level than at the RNA level. This result would be more convincing if the authors could show that the risk-associated SNPs at these loci are QTLs for the estProtein levels of these genes. If so, can it be demonstrated that the protein abundance has a mediating effect? Testing for QTL effects and mediating effects would more directly establish (or rule out) the connection from SNP to protein to cognition.

Response: Thank you for bringing up an important question regarding the utility of the data. The e/pQTL analysis usually needs to remove latent factors such as principal components for the transcriptome or proteome data. These latent factors represent cell composition, cellular compartments, trans effects from other genes, and the effects from environmental factors and batches. The removal of latent factors enriches local genetic effects and thus greatly improves the power to detect e/pQTLs (Stegle et al. 2010). Conversely, our protein prediction method does the opposite thing, adding latent factors to mRNA levels to predict its protein abundance, this essentially masks the cis-genetic effect. Therefore, the output from our prediction model is not suitable to investigate local genetic effects. To confirm this, we conducted pQTL analysis with the predicted protein data. We did not find strong pQTL effects as expected. The purpose of focusing on GWAS genes is to provide an implication where known AD genes are potentially dysregulated. We used AD GWAS as a source of AD genes as those genes are thought to be upstream factors for disease development. In addition to this, we are particularly interested in the alteration of AD GWAS genes in protein levels, as we didn't find strong evidence of differential expression at transcript levels in the previous study (Mostafavi et al. 2018).

Reference

- Stegle, Oliver, Leopold Parts, Richard Durbin, and John Winn. 2010. "A Bayesian Framework to Account for Complex Non-Genetic Factors in Gene Expression Levels Greatly Increases Power in eQTL Studies." *PLoS Computational Biology* 6 (5): e1000770.
- Mostafavi, Sara, Chris Gaiteri, Sarah E. Sullivan, Charles C. White, Shinya Tasaki, Jishu Xu, Mariko Taga, et al. 2018. "A Molecular Network of the Aging Human Brain Provides Insights into the Pathology and Cognitive Decline of Alzheimer's Disease." *Nature Neuroscience* 21 (6): 811–19.

4. In the pseudotime analysis (Fig. 4), as with the analyses in Fig. 3, I would find it very helpful to see comparable analyses to test associations of pseudotime with cognitive score, pathology score and diagnosis.

Response: We added association results for pathology score and diagnosis in Figure 6ab and for cognitive score in Figure S9. For all the metrics, estProtein-based pseudotime showed stronger associations than RNA-based pseudotime.

5. Connecting modules to GWAS. Fig. 5c. Stratified LDSC is not optimal for Gene Set Enrichment Analysis, as searching for enrichments in large regions around each gene in the network lacks specificity. Please confirm that results are robust when using a more standard tool for GWAS gene set analyses, such as MAGMA.

Response: Thank you for the insightful suggestion. We saw a good correlation between the result of LDSC and that of MAGMA (Figure R8). However, none of the modules passed the statistical significance in MAGMA. Because of the lack of robustness and the additional contents regarding model validation and interpretation in the revised manuscript, we decided to remove the analysis to link modules with GWAS from the paper.

Figure R8. Comparison of p-values from Magma and LDSC

Fig. 5d. *The PRS analysis was performed using an unusually small number of SNPs (just 159 SNPs). Typically, PRS analysis explains more variance and becomes more robust when thousands of SNPs are included in the prediction model. Can the authors demonstrate that this result is robust to the p-value threshold used in calculating the PRS?*

Response: We varied the p-value threshold for SNP inclusion in the PRS and noticed that the correlations of PRSs with a stringent threshold and ones with a relaxed threshold are unexpectedly low ($R \sim 0.15$) (Figure R9). The PRSs with a stringent threshold showed stronger associations with cognitive decline and AD diagnosis and the significance topped at p-value of 10^{-5} (Figure R10). To examine whether the stringent PRSs and the relaxed PRSs independently contribute to AD phenotypes, we ran additional linear models to both $\text{PRS}_{p \leq 10E-5}$ and $\text{PRS}_{p \leq 1}$. Interestingly, they were independently associated with cognitive decline and AD diagnosis (Table R1), indicating they represent different components of AD genetic susceptibility. From these results, we think that the use of $\text{PRS}_{p \leq 10E-5}$ is appropriate since it showed the strongest association with the AD phenotypes and the SNPs in the PRS include well-established AD loci. To our knowledge, the presence of two independent genetic components has not previously been detected. In summary, while the original finding is generally relevant to AD, given the additional findings, extensive characterizations for multiple PRSs may require even more elaboration around a secondary analysis. Therefore, we removed the PRS section from the paper entirely.

Figure R9. Pairwise correlation of PRSs with different p-value thresholds for SNP inclusion.

Figure R10. Associations of PRSs with cognitive decline and AD diagnosis.

Model	N	Term	P-value
(Cognitive decline)~PRS _{p<10E-5} + PRS _{all}	2,417	PRS _{p<10E-5}	3.03×10^{-23}
		PRS _{all}	2.33×10^{-14}
(AD diagnosis)~PRS _{p<10E-5} + PRS _{all} + age_death + sex + (years of education)	1,435	PRS _{p<10E-5}	1.16×10^{-11}
		PRS _{all}	9.30×10^{-9}

Table R1. Result of joint association model.

6. The article lacks validation studies for any of the proteins. Since the correlations of estProtein with observed proteomics signal were often quite low, some direct experimental validation of key findings seems essential. Which proteins to validate is the authors' choice but might include hub genes from early, middle, and late modules described in Fig. 5. The examples from these modules mentioned in the Discussion (line 411) are very interesting but would be more convincing with validation.

Response: As demonstrated by new results addressing this concern in Figure R2, the level of correlation is not a substantial concern for investigating disease effects via the predicted protein data. In addition, we actually have already provided two validation datasets, namely, the SRM study from the ROSMAP cohort and the MSBB study, and both clearly showed the improvement in aligning protein data and RNA-seq data with our method. Given the dual existing validations across cohorts and methodologies, an additional validation measurement is beyond the scope of this study as it requires a large-scale sample preparation for post-mortem tissues to obtain statistical significance. Nonetheless, to provide more evidence that our model indeed facilitates disease protein discovery, we conducted an additional analysis of the MSBB data. The MSBB cohort is completely independent of the ROSMAP cohort and technologies used for RNA-seq and protein data are also different, which makes the MSBB data very challenging validation data. We applied the model trained with the ROSMAP data to the MSBB RNA-seq data and obtained predicted protein data.

We followed the typical scenario for discovering disease-related proteins with RNA-seq data, where (i) run association analysis with the RNA-seq data, (ii) select candidates from differentially expressed genes, (iii) measure protein levels for those genes to validate they are differentially expressed at protein level. We used the CERAD score, neuropathologic diagnosis of AD, as a phenotype for this experiment. We picked the top N genes associated with the phenotype in RNA-seq data and the predicted protein data, respectively. Then we examined how many of them are differentially expressed with the actual protein data from the same individuals (Figure R11).

For instance, out of the top 100 significant genes in RNA data, 69 matched the direction of the association at the actual protein levels, and of these 23 showed p-value < 0.05. In contrast, 80 out of the top 100 genes in the predicted protein data matched the direction of the association with the actual protein data and 36 showed p-value < 0.05. For reference, we also selected genes randomly, resulting in 11% of success rate. Our prediction model improved the success probability of disease protein discovery 1.6-fold over the RNA-based gene selection and 3.4-fold over random selection. This demonstrates that our approach significantly improves traditional disease protein discovery processes with the power of deep learning. This result is included in the revised manuscript as Figure 4g.

Figure R11. Success rate of disease protein discovery based on mRNA and estimated protein. The clei2block model trained with ROSMAP data was applied to MSBB data, which is entirely independent of the ROSMAP cohort. AD-related genes identified from the estimated protein data are more likely validated with the actual proteome data than those from raw RNA expression.

Minor points

1. Figure 1 is really a graphical abstract. I recommend that this be moved to the supplement.

Response: We added sample size and the clei2block architecture in Figure 1, which we believe made it more than just a graphical abstract.

2. Much of the introduction is devoted to a summary of results that would be better suited for the first paragraph of the discussion or removed.

Response: Thank you for bringing this to our attention. We reduced the summary of our findings from the abstract.

3. There are some missing details regarding the ensemble averaging procedure. How was averaging done? Were all of the underlying models assigned equal weight? Etc.

Response: We just took the average of all outputs with equal weight. We clarified this procedure in the method section.

4. Please provide exact p-values on figure 2 (not $p < 2.2e-16$).

Response: We added actual p-values to all figures in the revision.

5. I find the annotation of “medial frontal” cortex to be vague. The dissections were for much more precise sub-regions. For instance, Yu et al. (2020) in JAMA Psychiatry

describes the proteomics samples as being from DLPFC. Please state exactly which sub-regions were used and combined.

Response: The TMT proteomics data is the same as one reported in Yu et al. (2020) in JAMA Psychiatry. Therefore, they are from DLPFC to be more specific. We revised region name.

6. On Fig. 3c, I find the y-axis of this figure confusing relative to the text. APP has a negative value for estProtein, and PICALM has a positive value. But the text indicates that APP is associated with an increased rate of cognitive decline. I think I can piece together what is being shown, but it's not clearly stated. An additional 1-2 sentences in the figure legend are needed.

Response: Cognitive decline is actually a slope of cognitive score over time. So the negative cognitive decline means that cognitive score is declining. While somewhat challenging to interpret, this has been the convention in dozens (all) of previous papers. Nevertheless, at your suggestion, we changed the sign of cognitive decline in the revised manuscript so that higher cognitive decline indicates faster progression of dementia.

Seth Ament

Reviewer #2 (Expertise: deep learning, OMICs data analysis, RNASeq):

Remarks to authors:

The manuscript "Protein systems in Alzheimer's dementia, via deep neural network analysis of RNA expression" by Tasaki et al proposes a deep neural network model to predict protein abundance from RNA expression. The authors present a few examples that show promise. In particular, the authors show the application of their predictive model to identify proteomic signatures associated with ADRD but I have major concerns about this paper as well. I found it difficult to understand the experiments explained by authors (i.e. what are train/valid/test splits) and generally hard to follow. Most of the analyses are based on in-silico predicted data which I believe is hard to evaluate without real ground truth and more careful evaluations. Additionally, the importance of the different modules of the pipeline is not justified through an ablation study to show why those pieces improve the prediction. Finally, the metrics make it hard to understand how well or poorly their method is performing.

Response: We apologize that our description of the experiments was not clear enough, and have expanded it in the revised text with Figure S1 and S2. In brief, to train the model, we split the samples into ten folds, taking care to balance cognitive status and brain pathologies. Then, for each fold, we removed the fold from the data and used the remaining data (90% of data) to train the model. Within the training samples, 10% of them were used as validation data to monitor learning progress and select the model that worked best for the validation data during the iteration. The trained model was applied to the holdout testing data so that we obtained predicted protein data for the holdout samples. We repeated this process for each of ten holds,

resulting in obtaining 10 models trained with different training samples and holdout testing data. To calculate overall prediction accuracy, predictions for the holdout samples were concatenated after scaling within each hold and then contrasted with actual proteome abundance. We want to emphasize that predicted protein data for the holdout samples was generated based on models trained without using any holdout samples, which ensures no information leakage from the holdout samples. We expanded the method section to describe this experimental setting in more detail.

Major points:

1. While the prediction of protein abundance from RNA has been shown feasible with reasonable accuracy (Buccitelli and Selbach 2020), Fortelny et al found that this prediction also works when protein-to-mRNA ratios are calculated from randomly shuffled or identical mRNA levels (Fortelny et al. 2017). I would like to see further experiments from authors to demonstrate their predictions are meaningful and such results are not possible with random or identical mRNA levels.

Response: An important distinction between our prediction model and the method proposed by Wilhelm et al (2014) that is criticized by Fortelny et al. (2017) is that Wilhelm et al. proposed a method to predict differences in protein abundance between proteins in each sample. Thus, their metric is an across-gene correlation. In contrast, our model predicts differences in protein abundance between samples and the metric is an across-sample correlation for each gene. Predicting protein abundance between genes is not the focus of this study, as our primary goal is to understand protein systems associated with phenotypes across individuals. One additional difference in our approach and that of Wilhelm et al. is that a key variable in their method is a translation rate which is estimated from protein data they intended to predict. This essentially means that they used protein data as an input of the model to predict the protein data itself. This caused significant information leakage and thus additional information from mRNA levels are almost meaningless, as the model still produced a reasonable prediction with random or identical mRNA levels, as Fortelny et al. rightly point out. We found that criticism to be on point, and therefore from the start set up our model to NOT use any information from the protein data, but only RNA data as an input. Accordingly, the model with identical RNA data generates identical protein levels and the correlation between predicted and actual proteins cannot be calculated. Also, as expected, and unlike the Wilhelm model, our model with random RNA input data generates random protein levels.

2. The model design and existence of specific components are not well justified, for example, it is unclear that pre-mRNA and UTR length would increase perf a lot. Authors should indicate correlations between each of those and proteins. Or why do we need scGen there at all? Generally, It would be beneficial for the readers to see careful and extensive ablation studies to show the relevance of each part of the model.

Response: We very much appreciate your thoughtful comment. Our strategy was to extract information from RNA-seq data without any strong hypothesis, in order to explore the key

factors which explain protein abundance. Although we have provided the predictive contribution for each variable, the collective contribution of each data type could be more informative as the reviewer suggested. To evaluate this, we conducted an extensive ablation study where we trained the model with the removal of each data type and compared the performance with the model trained with a full dataset. Removing any of the data types reduced the predictive performance (**Figure R12**). The most influential data type is mRNA in the linear-regression module, which biologically makes sense as mRNA levels of the gene should have a direct effect on its protein levels.

To evaluate the necessity of the scGen component in the clei2block model, we compared three models: (i) clei2block that is a combination of the scGen and LR (linear regression) modules, (ii) the NN-linear that is a combination of a fully connected layer and the LR module, (iii) the scGen module. The clei2block performed the best followed by the scGen and then the NN-linear model (**Figure R13**). The difference between clei2block and NN-linear is that replacing the scGen module that executes non-linear transformation of input variables with a fully connected layer that operates linear transformation. This indicates that non-linear transformation is critical to achieving higher predictive accuracy. We have included these results in the revised manuscript as Figure 2c and 3a.

Figure R12. Ablation experiment to identify key data types. The clei2block model was trained with the removal of entire input variables of each data type as indicated in the heatmap. The average Pearson’s correlations between predicted and actual protein levels for each model are indicated in the bar graph on the right.

Figure R13. Performance comparison of different neural net architectures. The clei2block was compared with the model without the scGen component and the scGen models.

3. 90-91: *“To mitigate the discrepancy, we developed a predictive model of the proteome solely based on transcriptome data”, I found this part negating the whole section if the correlation is already low and not explained in the RNA data how could such data predict the protein level explaining such variances?*

Response: We apologize that our description sounded contradictory. The issue we intended to raise is that the correlation between protein levels and mRNA levels for the protein across individuals is low. This is the simplest prediction model as it is based on a single variable. The approach we proposed does not rely on a single gene but instead uses multiple genes to predict protein levels. To clarify this point, we revised the description as below.

“To mitigate the discrepancy of protein levels and corresponding mRNA levels, we developed a predictive model of the proteome based on multiple variables extracted from transcriptome data.”

4. It would be beneficial if authors add additional baseline or existing methods to compare their model against them to further benchmark the model.

Response: We appreciate this suggestion. We added a tree-based method and neural network model without the encoder part (**Figure R14**). This additional experiment demonstrates that the clei2block model outperforms the standard baseline models. We have included the result as Figure 2b.

Figure R14. Performance comparison of clei2block with Elastic-net and CatBoost. We trained the Elastic-net and CatBoost models for each protein separately using the same data and sample splits with those of the clei2block model. Because training the Elastic-net model took a significantly long time with a large number of predictors, we focused on the six submodels that take mRNA-PCs, pre-mRNA-PCs, or UTR-PCs as predictors in this comparison. A blue diamond indicates the performance of ensemble prediction.

Minor points:

1. Figure 3c, can you investigate the effect size here?

Response: Thank you for your suggestion. We have added a proportion of variance explained in Figure S8.

2. I looked into the GitHub repository trying to reproduce or generate some of the results of the paper but were unable to do so. It would necessary to have some tutorials for the users and allow them to reproduce the results of the paper (or at least some part of it)

Response: We have added a tutorial for our model with sample data in the GitHub repository. Since the size of the actual model is too large to deposit to GitHub, we have created another repository in Synapse for the trained model (<http://dx.doi.org/10.7303/syn23624037>) and anonymized data. The ROSMAP and MSBB data are controlled-access data, which prohibits us from putting the data into a totally open sharing system. However, the access is readily granted through the RADC Resource Sharing Hub at www.radc.rush.edu or the AD Knowledge Portal (<https://adknowledgeportal.org>).

3. Figure 5D and S12: authors need to report these associations for RNA as well.

Response: We added the association of RNA modules and cognitive decline in Figure S11. We decided to remove the PRS analysis from the revised paper, as it was becoming very involved. However, for your reference, we provide the association for RNA modules here (**Figure R15**).

The associations are similar but less significant than those of estimated proteins, which is consistent with the other analysis.

Figure R15. Associations of protein modules with cognitive decline and AD-PRS. Module expression levels based on the predicted proteomes from the DLPFC (n=1,192) were regressed with cognitive decline or AD-PRS. For reference, we also computed module expression levels of RNA expression given module definition based on the predicted proteomes. Negative log10 of the p-value with the sign of its association to cognitive decline is indicated in the bar plot.

4. Why have authors leveraged a trajectory inference method for scRNA-seq data for RNAbulk and proteomic data where a lot of assumptions might not hold?

Response: We are motivated to utilize this approach, given insightful findings from several other papers inside and outside the AD field, all using bulk RNA-seq data. For instance, Iturria-Medina et al (2019) applied it to blood and postmortem brain bulk RNA-seq to model the progression of neurodegenerative disease. Mukherjee et al (2020) also showed that a trajectory method can be applied to bulk RNA-seq to extract meaningful molecular-based stages of AD.

References

Iturria-Medina, Y., Khan, A. F., Adewale, Q., Shirazi, A. H. & Alzheimer's Disease Neuroimaging Initiative. Blood and brain gene expression trajectories mirror neuropathology and clinical deterioration in neurodegeneration. *Brain* **143**, 661–673 (2020).

Mukherjee, S. *et al.* Molecular estimation of neurodegeneration pseudotime in older brains. 686824 (2020) doi:10.1101/686824.

References

Buccitelli, Christopher, and Matthias Selbach. 2020. "mRNAs, Proteins and the Emerging Principles of Gene Expression Control." *Nature Reviews. Genetics* 21 (10): 630–44.

Fortelny, Nikolaus, Christopher M. Overall, Paul Pavlidis, and Gabriela V. Cohen Freue. 2017. "Can We Predict Protein from mRNA Levels?" *Nature* 547 (7664): E19–20.

Reviewer #3 (Expertise: Therapeutics for Alzheimer's disease):

This is a very important breakthrough study to address a major challenge that faced the field of AD research, especially with regard to the translational aspect of drug discovery. There has been a tremendous effort to generate RNA transcriptomics from post-mortem brain tissues and compare that to transgene mice models to better understand the trajectory of the AD pathology. However proteins carry out the actual work and are closer to the observed phenotype, but technical challenges have made large-scale proteomic studies lag behind their RNA-based -omics counterparts.

The authors use an innovative deep learning technique to develop a "protein-like" system analysis, based on a training set of concordant proteomic and transcriptomic data of a relative small subset of subjects in the ROSMAP database and apply this to the largest available database in the field. They show a limited overlap between the transcriptomics and "proteomic like" based prediction models and that the proteomic-like prediction model has a closer relationship with actual clinical phenotype such as hippocampal atrophy and cognitive decline.

The authors address very well the shortcomings of their approach in putative generalizability (although to their credit they use different datasets to validate their findings). Interestingly they identify some solutions to address these challenges with new datasets and patient populations.

The paper is well written and the experimental details provided in sufficient detail.

A few minor comments to add to the discussion:

1. It might be of interest to mention and contrast to other methods for predicting tissue-specific translation using large scale proteomics and transcriptomic data resources such as non-neural network approaches (e.g. Edfors 2016

<https://www.ncbi.nlm.nih.gov/pmc/articles/PMC5081484/>).

Response: Thank you for the words of encouragement and also for providing us an opportunity to clarify our method. Their main finding of Edfors 2016 is that the RNA-to-protein (RTP) ratio for

each gene is preserved across tissues. Therefore by multiplying RNA copy number with the RTP ratio, you can predict protein copy number. In Figure 5B, they showed RTP-based prediction outperformed the original RNA. However, it is important to point out that the metric used in Figure 5B is a correlation of protein copy numbers across different genes in each tissue. In contrast, our model predicts differences in protein abundance between samples and the metric is an across-sample correlation for each gene. The RTP-based method does not predict across sample differences of protein, because the fixed RTP ratio is multiplied with RNA copy numbers from different tissues/samples. The estimated protein copy numbers across tissues are still perfectly correlated with the original RNA copy numbers for each gene. We expanded the discussion to clarify that our model predicts protein abundance between samples rather than those between proteins.

2. An interesting finding is the appearance of APP and PICALM in the protein-like prediction model but not in the RNAseq model, a fact the authors mention in their introduction. It would be of interest to discuss how these new proteomic markers would appear after the deep-learning approach. On the other hand, APOE the most robust genetic risk factor for AD has only a weak effect in the proteomic-like system analysis.

Response: Thank you for raising this important point. We have analyzed the model to understand what biological systems drive the associations of APP and PICALM with cognitive decline. We found that the 79th principal component of mRNAs (PC79) affected both proteins. PC79 is associated with the replication fork but the association is not strong. Another key predictor of APP was PC1, which is strongly associated with neuronal genes. This might indicate that APP protein is predicted through the number of neurons or molecular state changes in the neuronal population. We did not find statistical evidence of differential expression of APOE at protein levels. This might be due to the limitation of bulk omics data as recent single-nucleus RNA-seq data from our cohort suggested that APOE is differentially expressed in AD in microglia. With the advance of proteomics technology, we might be able to measure proteins at single-cell resolution. Alternatively, it might be possible to extend our model to single-nucleus RNA-seq data and predict cell-specific proteomes computationally. We expect that experimental or computational advancement will identify cell types or regions where APOE protein is dysregulated in AD brain.

3. Post-translational modifications (phosphorylation, acetylation, glycation, truncation, etc) of many AD-related proteins drive the progression, which is obviously beyond the scope of this report.

Response: Thank you for your comment. As the reviewer mentioned, predicting post-translational modifications is beyond the scope of this paper. We hope to measure some of the key post-translational modifications at a proteome-wide scale in the ROSMAP cohort and evaluate if the clei2block model can predict them from other omics data.

4. Correlation or association is not per se causation; the biology is often more complex than a linear relation between expression levels and clinical phenotype (as has been

painfully demonstrated in the amyloid modulating clinical trials). These approaches do not obviate the need for further biological research to really use this as a target selection tool.

Response: We totally agree that this method is not designed to distinguish cause and effect. Our primary goal is to screen important genes based on RNA-seq data for further validation study that can investigate the causality of those genes against AD phenotypes.

Hugo Geerts, Certara

Reviewer #4 (Expertise: OMICs data integration):

Summary

Tasaki and colleagues implemented a deep neural net for the estimation of protein abundances from the mRNA abundance, pre-mRNA abundance, and 3'UTR length from the transcriptome data as input. The in silico translated data are further analyzed in three different ways, namely co-expression network analysis, association with cognitive decline, and pseudotime analysis that reflect the dementia progression. Despite seemingly interpretable results, it won't be a stretch to say that the validity of all reported findings hinges on successful prediction of protein abundance values. The authors make a case that the predicted protein abundance data improve the aforementioned systems-level inference compared to an analysis where mRNA abundance are regarded as proxy of protein abundance. From my point of view, however, these downstream implications are indirect metrics of success and do not validate the accuracy of proteome prediction. The detailed comments are as follows.

Major comments

1. The straightforward evaluation of the prediction performance should be based on the correlation between estimated relative protein abundance and experimentally measured protein abundance for individual proteins. In this sense, I think it's important to present the distribution of Pearson correlations between estimated proteome and targeted proteomics data in a histogram in the training data (TMT / N=384) as well as test data (against targeted proteomics data) in Figure 2. Through Figures 2a-2c, the authors use the false discovery rate for the hypothesis testing of H_0 : correlation = 0 and H_A : correlation \neq 0 as the evidence of improvement over the surrogate (mRNA), where the correlations are computed between predicted protein values and experimentally measured protein abundance, or between mRNA abundance and experimentally measured protein abundance. Using statistical significance measures of the correlations with actual measurements is equivalent to considering association as causality or prediction. Of course the authors have trained a prediction model in a way to increase the correlation with measured proteins, so the correlation increased. But how good or bad are the correlations?

2. In fact, I am sorry to say that I cannot lend much support to the prediction accuracy, which is briefly mentioned and undermines the credibility of the results. On page 4, line 110, it is stated that the average correlation between estimated protein abundance and TMT proteomics (cross-validated) for all proteins was 0.18. This means that you must have had a large number of proteins with nominal correlations close to zero or even negative. The first thing that comes to my mind is that you may need to filter some poorly predicted proteins from the estimated data.

Response: Thank you for your comments 1 and 2. We address both comments here as they are closely related. First, we added violin plots of Pearson's correlations in Figure 2 of the revised manuscript. Second, we examined the theoretical limit of prediction accuracy. To do this, we compared TMT proteomics and SRM proteomics data measured from the same brain region, in the same individuals (N=384). The average Pearson's correlation is 0.2. This suggests that achieving prediction accuracy of greater than 0.2 is very challenging for our data. Considering the difficulty of this task, we believe the performance of the clei2block model is reasonably high. The most important thing we would like to emphasize here is that this seemingly low correlation does not limit the ability to investigate disease effects with our data. We discussed this point in the response to the first comment of reviewer 1 in detail. Regarding the protein selection for the downstream analysis, we used proteins with Pearson's correlation greater than 0.1, which ensures results are based on high-quality prediction.

3. Here is a more philosophical comment. If the proteome could be robustly predicted from the transcriptome data, that will be revolutionary. The irony here is that the authors describe the motivation of the work as the poor correlation between the transcriptome and the proteome, yet they infer the proteome from the transcriptome. Essentially, at the tissue level, the authors must be inferring that the amalgamation of all mRNA transcripts somehow yields good predictive information of quantitative levels of individual proteins, and this mechanism can be learned from a collection of a few hundreds of brain tissue samples.

Response: We agree with your summation of the model. To reiterate, one of the benefits of neural networks is flexibility in the model structure. We can easily arbitrarily combine multiple inputs and jointly optimize them. In our case, we can conduct a non-linear transformation of transcriptome data to proteome data (encoder part) and then add gene-specific mRNA levels at the last step (LR part). The benefit of the encoder part is that parameters are shared between different genes, which means neural networks can find more generalized latent factors by leveraging the information across different genes. This does help overcome the apparently low gene-protein correlation, as the information needed to relate them does exist in a disguised fashion in the transcriptome, in a manner that is not easily utilized by traditional prediction models. Accordingly, all the inputs need to be combined and processed in the same way, and the model has to be trained for each protein independently.

I have a very hard time believing this. Deep neural networks are most successful in computer vision and image processing, where the data structure are values on grid coordinates on a lattice (e.g. pixels with a fixed number of neighbors), where the task is

to identify certain patterns or types of objects. Even for these “simpler” tasks, convolutional networks or other types of deep learning methods often need to be trained on hundreds of thousands, if not millions, of picture frames to attain reasonably error-free performance. Protein data are not on lattice grids and there is no regularity. Translation of each and every protein is governed by a different combinations of factors and cellular context, and therefore one can surmise that the protein abundance is likely a function of many biological inputs, not just the abundance of the template (mRNA) and the post-transcriptional and translation machinery in a specific tissue type.

Response: While some neural network studies have the luxury of utilizing millions of images etc, there are also dozens of examples where deep neural networks are used for areas outside of vision and imaging. Especially in biology, there are many successful examples in extracting meaningful information from transcriptome data. In fact, the scGen model from which our model is inspired is a deep neural model for transcriptome data. So even if you have doubts about our particular model - which we hope we have addressed entirely away in our response - the question of if a limited number of transcriptome samples can be harnessed with neural networks has already been settled by previous talented researchers. To the question of the performance of our model, we conducted an extensive comparison with non-neural network-based approaches. None of them showed comparable performance with the clei2block model.

Do authors believe that they have sufficient training data to account for such complex mechanisms and claim that their deep neural net is not overtrained in their own data?

Response: This is really the meat of your “philosophical” question and we appreciate the opportunity to provide concrete results on this. We investigated the relationship between sample size and prediction accuracy by subsampling training data. We found that performance is almost saturated with our current sample size (Figure S3). This is likely due to the measurement noise and this noise cannot be predicted by the other independent data using any prediction models if the noise is independent. This indicates that our model is properly trained and the sample size is sufficient.

If you can prove that you have very good prediction results (e.g. $R^2 = 0.9$ for thousands of proteins) with the presented data, this will be not an ordinary feat.

Response: As we showed with the actual protein data, the practical limit of the accuracy is about 0.2, and this level is still sufficient for important phenotypic results on Alzheimer’s, as we’ve shown in the paper, and new results shown to reviewers

Minor comments

1. In line 27-29, I’m pretty sure we are way past the time when transcriptomic data are the best background to look for drug targets.

Response: While there are many molecular sources, we would like to have available in thousands of subjects, the main data that is actually available and at the core of most of the projects in the largest drug discovery consortium for Alzheimer’s disease in the US is still RNA-

seq. While it has limitations, it's also a mature technology in terms of both measurement and data analysis, and thus can be applied to large-scale clinical samples. Indeed, we have been successful to nominate drug targets from RNA-seq data (Mostafavi et al. 2018; Yu et al. 2018; Mathys et al. 2019). Of course, we are working to push beyond this, but the global proteomics technology is just becoming available for large-scale clinical samples and the sample size is far less than RNA-seq. However, with the limited protein data, we realized there are significant differences between RNA abundance and its protein levels in our brain data and protein abundance showed stronger associations with AD traits than RNA levels. Therefore, we expect that proteome will be used in the field of AD more frequently. At the same time, it is very important to consider how we can leverage the large-scale transcriptome data we have collected. Also, some new technologies such as single-nucleus RNA-seq and spatial transcriptomics are only available for RNA levels. Thus, it is worthwhile to start considering how we can extract protein information from those new RNA-seq technologies. Our study provides a possibility of predicting protein abundance from RNA-seq data, which could be extended to other advanced RNA-seq data.

References

- Mostafavi, Sara, Chris Gaiteri, Sarah E. Sullivan, Charles C. White, Shinya Tasaki, Jishu Xu, Mariko Taga, et al. 2018. "A Molecular Network of the Aging Human Brain Provides Insights into the Pathology and Cognitive Decline of Alzheimer's Disease." *Nature Neuroscience* 21 (6): 811–19.
- Yu, Lei, Vladislav A. Petyuk, Chris Gaiteri, Sara Mostafavi, Tracy Young-Pearse, Raj C. Shah, Aron S. Buchman, et al. 2018. "Targeted Brain Proteomics Uncover Multiple Pathways to Alzheimer's Dementia." *Annals of Neurology* 84 (1): 78–88.
- Mathys, Hansruedi, Jose Davila-Velderrain, Zhuyu Peng, Fan Gao, Shahin Mohammadi, Jennie Z. Young, Madhvi Menon, et al. 2019. "Single-Cell Transcriptomic Analysis of Alzheimer's Disease." *Nature* 570 (7761): 332–37.

2. In Figure 1, it will be helpful to have more detailed information indicating what part of the samples were used as input to train the deep neural net (transcriptome + global proteome), and what other part of the samples with what data (e.g. transcriptome, validation by targeted proteomic data), with sample sizes indicated in appropriate places.

Response: Thank you. We have revised Figure 1 to reflect the reviewer's suggestion.

3. It is unclear why the authors chose regression with elastic net penalty as the only competing model. In predicting the abundance of protein A, was it possible for the mRNA features of the same gene to be omitted from the prediction model as a result of variable selection? Could you have chosen more sophisticated non-linear predictors like SVM with radial basis or tree-based gradient boosting / random forest?

Response: Thank you for your suggestion. First, it is possible that the mRNA feature of the same gene would be omitted from the prediction model as we did not put any restrictions on the variable selection of the elastic net. We agreed with the reviewer that a comparison with a non-linear model would be informative. We built additional models with tree-based gradient boosting. The performance of gradient boosting was inferior to a simple elastic net model. This might be because hyperparameters need to be highly tuned to get optimal performance. However, the computation time for the gradient boosting is significantly longer than elastic net or neural networks, which is prohibitive to complete through hyperparameters tuning in a reasonable time. In this comparison, we only tuned the number of trees in the model. We described this limitation in the revised paper.

4. Figure S3. What test statistics are you using? Can you add the info to the figure legend?

Response: We used t-statistics in this comparison and have added a description to the figure legend.

5. In lines 181-191, can you provide more detailed account to give insights as to how post-transcriptional regulation and translation machinery genes contributed to the prediction of abundance for individual proteins? Or do we just know that those genes had higher feature importance scores / causality scores?

Response: This is simply based on feature importance scores. Specifically, we conducted gene-set enrichment analysis for the genes with higher importance scores and visualized enriched gene ontology terms.

6. It was observed in the paper that amyloid beta precursor (APP) gene itself was not differentially expressed in AD cases at the mRNA level, but the protein was (in TMT measured portion as well as in the estimated proteome). Were you able to explain what data features contributed the most to the elevated protein level of APP? Delineating this would be revolutionary, whether the increased APP level is driven by increased translation, or simply a defect in the neuronal clearance of amyloid beta.

Response: Thank you for bringing up an important question. We conducted additional analysis to understand features deriving the elevated protein level of APP. Genes expressing neuronal cells are the major predictors for APP elevation. Although the model does not suggest a causal relationship between predictors and outcome, this might reflect the changes in the number of neurons or molecular state of neurons caused by amyloid deposition. We have added this result in the revised manuscript as Figure 5d.

7. A lot of details regarding proteomics experiments are lacking in the Methods section. I recommend the authors to include detailed supplementary information regarding all experimental parameters (both details of TMT-labeling-based global proteome analysis and the transitions / assay info of the SRM). The presence of supplementary information

may allow the authors to reduce word counts by moving the description of clei2block to the Supp Info as well.

Response: Our study is based on the TMT and SRM data already published by other researchers, and we downloaded these data from the Synapse repository. We believe the papers we cited and the descriptions in the Synapse repository will provide sufficient information for the proteomic data generation processes.

Reviewer #5 (Expertise: TMT proteomics):

The authors have developed a deep neural network model to predict protein abundance from RNA expression in aged brains. This would be of great interest to the scientific community, not only for Alzheimer's and neurodegenerative disease research but also for advancing the compatibility of transcriptomics and proteomics analyses. I believe the work should be published upon addressing my below concerns.

In focusing on the proteomics experiments, the already published TMT experiments appear to be well thought out and performed in a way consistent with how they are generally performed with the proteomics field. My concern is regarding the SRM experiments, as there is little mention to how they were performed in this work, and the references cited in the manuscript for the SRM experiments do not appear to involve SRM experiments at all. It is unclear if I am missing something or if the authors referenced the wrong papers.

Additionally, because of the potential interest from a broad audience, I believe a broader explanation of some items in the manuscript should be added. For example, the abbreviation "GWAS" should be explained.

Response: Thank you for your encouraging comments and suggestions. We reported the detailed methods for SRM experiments in (Yu et al. 2018). We revised the method section to clarify that our SRM experiment is based on the report previously published. Also, we have expanded the abbreviations as much as possible.

Reference

Yu, Lei, Vladislav A. Petyuk, Chris Gaiteri, Sara Mostafavi, Tracy Young-Pearse, Raj C. Shah, Aron S. Buchman, et al. 2018. "Targeted Brain Proteomics Uncover Multiple Pathways to Alzheimer's Dementia." *Annals of Neurology* 84 (1): 78–88.

Reviewer #6 (Expertise: Statistics, longitudinal data analysis):

The paper "Protein systems in Alzheimer's dementia, via deep neural network analysis of RNA expression" presents a new approach to predict protein abundance from RNA expression, which is validated on multiple datasets. Three applications further demonstrate certain advantages of the predicted protein levels over mRNA signals.

However, the current version of the paper does not provide enough explanation to the mechanism or enough benchmarking. The comments are listed below:

1. The clei2block extracts low dimensional lateral factors from transcriptomes for model building, and the authors have tracked down the origin of the predictive accuracy from the protein specific input such as mRNA, pre-mRNA and UTR length and encoder input. However, the authors have not investigated the decoder output, which is used for linear model building as downstream input. Please provide more details on how the decoder output is different from the encoder input, how important is the scGen module in the clei2block pipeline?

2. The authors have made comparison with the Elastic-net model. Please compare clei2block with the model by only using linear regression module with the encoder input and protein specific input.

Response: We very much appreciate your thoughtful comment. We believe that comments #1 and #2 are closely related and it's better to respond to them together here. The scGen module operates a non-linear transformation of the encoder input. We agree with the reviewer that gauging the contribution of the decoder output from the scGen module would be very informative to understand how non-linear transformation benefited the model. To do so, we followed the reviewer's 2nd idea where we compared the clei2block model with a model only based on the linear regression module. The result is shown in **Figure R16**. The model without the scGen module (NN-linear) underperformed against the clei2block model about 21% in terms of average correlation. Even more, the performance of the NN-linear model is worse than the scGen-only model. This suggests that non-linear transformation by the scGen module is essential in our model.

Figure R16. Performance comparison of different neural net architectures. The clei2block was compared with the model without the scGen component and the scGen models.

3. As the authors have mentioned that genes projecting to influential PCs were enriched for translational initiation and mRNA processing, such as ribosomal proteins, translation

initiation factors, and splicing factors, which is rather general and not protein specific. Would these PCs introduce false positive in prediction? Please discuss this.

Response: We assessed the importance of PCs based on their average contributions across proteins. Therefore, if PCs contribute to the specific proteins, those PCs won't be picked up as influential PCs because they just impacted a small fraction of prediction accuracy overall. In light of this, it follows that influential PCs are involved in broad or fundamental biological processes such as protein translation, which does not mean they would necessarily introduce false positives in prediction.

4. The authors mentioned that “We found 3826 proteins associated with cognitive decline (Bonferroni-corrected $p < 0.05$) (Figure 3a and Table S4), which is 15-fold more than those identified from actual proteome data of 384 samples and 2.4-fold more than those from transcriptome data of the same 1,192 samples”. Please explain why the number of predicted proteins is much more than the actual proteome data in terms of association with cognitive decline. The more the better? Or is the actual proteome data unreliable?

Response: You raise a point in which we are also very interested. There would be a couple of reasons why we found more proteins associated with AD phenotypes. First, the sample size for the predicted proteome (N=1,192) is greater than the actual proteome data (N=384), as we converted large-scale RNA-seq data to proteome data. Second, the clei2block model potentially reduces the measurement noise in data. The reason for that is when the model is properly trained, it will not be able to learn noise that is independent between protein and mRNA measurements and between different genes. Thus, the predicted protein should have less noise. If this is the case, it may be one source of the stronger statistical significance we observe with AD phenotypes.

5. The authors evaluate the accuracy of clei2block by comparing the number of mRNAs and estimated proteins positively correlated with actual protein abundance, which is not straightforward. Please provide more details on prediction errors compared with the actual protein abundance.

Response: Thank you for your suggestion. We have added violin plots to show the distribution of correlation between actual and predicted proteins.

6. The authors should address f proteomics and highlight the advantages of transcriptome data over proteomics in the introduction section.

Response: We very much appreciate your thoughtful comment. We have revised the introduction to emphasize this point.

7. The authors mentioned that “Because training the Elastic-net model took a significantly long time with a large number of predictors”, so how long does clei2block

take to process the datasets in the manuscript. The author can address the computation efficiency of clei2block.

Response: We agree that this is practically very important. To train the model with 5,000 variables in the encoder input, the Elastic-net took around 9 seconds per protein using a single core of Intel(R) Xeon(R) CPU @ 2.30GHz, and thus 20 hours for all 8,391 proteins. In addition to this, the 10-fold-cross validation experiment requires 10 times more of this, which is 200 hours of computing time in total. For the CatBoost model, which we included in the revised manuscript, it took around 435 seconds per protein (1000 hours for all proteins) and thus will require more than 10,000 hours of computation in total. Although the CatBoost offered GPU-based training, it did not improve the training speed for our data. On the other hand, the clei2block model only took 9 minutes to train and thus 1.5 hours in total using the same CPU with an NVIDIA Tesla T4.

8. The clei2block can be used to identify ADRD-associated proteins with negligible changes at the transcript-level. For these proteins, which factors contribute mostly within the input? Please provide statistical analysis or references on how these proteins are influenced by the factors, such as ribosomal proteins, translation initiation factors, and splicing factors.

Response: We conducted an in-depth analysis to elucidate what biological systems contribute to the associations of AD genes with AD diagnosis, cognitive decline, and global AD pathology. We found that immune gene signature affected the associations the most, followed by ribosomal proteins, nuclear speckles, and neuronal genes (**Figure R17**). Although the model does not clarify whether these factors are indeed driving protein changes or vice versa, these factors might be a good starting point to investigate genes driving protein changes in AD brain. We appreciate your suggesting this investigation, and have included this result as Figure 5dc.

Figure R17. Influential predictors for the protein-trait association. The predictive contribution of each variable in the encoder inputs of mRNA-PCs was estimated using GradientExplainer. For each protein-trait pair, we selected the top two PCs, indicated by an asterisk, that strengthened their association. The color of the heatmap represents the

contribution of each PC to the protein variance explained by trait in percentage. Overall contributions of PCs to the associations across AD GWAS proteins and the traits. Representative gene ontologies enriched for genes contributing to the influential PCs were described in the bar plot.

9. In Figure S15, where is the “Ensemble prediction” of scGen? Please provide more information.

Response: We added an ensemble prediction of the scGen in Figure 2b. The ensemble prediction improved the prediction accuracy of the scGen model but it's not comparable with that of the clei2block.

10. In Figure 3a, the number of genes whose expression are associated with cognitive decline in AC dataset is much larger than the PCG dataset, while the number of genes whose estProteins are associated with cognitive decline in AC dataset is smaller than the PCG dataset. Is it due to batch effect? Please discuss this counterfactual phenomenon.

Response: We checked this phenomenon in different AD-related traits. With regard to AD diagnosis and global AD pathology, PCG has a larger number of differentially expressed genes (DEGs) than AC in both estProtein and mRNA. It is difficult to track down the exact reason for the reversal of the number of DEGs for cognitive decline but unknown confounding factors might happen to weaken the relation to cognitive decline or the sample size is not sufficient for robust identification of DEGs.

11. In the Fig 3c, it is suggested to place the gene symbol in the vertical axis and signed -log₁₀(P-value) in the horizontal axis, making it more clear to readers.

Response: Agreed! We switched the y- and x- axes.

12. In the Fig 3c, the gene ADAM10 is significantly associated with cognitive decline, while its estProtein is not. Please provide some possible explanations.

Response: Thank you for the question. Indeed, we often see an inconsistency between mRNA and protein, which is why we initiated this research. For instance, with the 384 ROSMAP samples where both mRNA and protein are available, there are 65 genes differentially expressed at mRNA levels. Out of 65, protein levels of only 2 are differentially expressed. We think that there are many reasons for this discrepancy. One of the reasons is the issue of statistical power. Alzheimer's disease is very heterogeneous and collecting clinical samples is not well controlled compared to in-vitro experiments. Therefore the power of the study is limited even though we have hundreds of samples. There are many possible biological reasons as well. For instance, to translate protein, mRNA needs to be at a certain cellular location. However, sometimes mRNA is translocated at the places where translational activity is low, such as certain types of RNA granule or extracellular regions. Also, post-transcriptional modifications such as m6A can alter the translatability of mRNA. In addition to this, the activity of protein translation and degradation would be different for each individual, which also leads to

inconsistency between mRNA and protein levels. It would be great if the model can provide some hints about the mechanisms. However, the model does not distinguish between causal relationships and correlations.

Reviewers' Comments:

Reviewer #1:

Remarks to the Author:

The authors have provided a thorough response to my concerns from the first submission, added numerous additional analyses that improve the paper, and removed a few that proved less robust. Altogether, I think this is a fascinating and important investigation, which I am hopeful will provide useful insights for understanding the pathophysiology of AD.

Reviewer #2:

Remarks to the Author:

The authors did a good job and addressed all my comments. I have no further comments.

Reviewer #3:

Remarks to the Author:

The authors have satisfactorily addressed my comments. The paper has substantially improved and is of high scientific value.

Reviewer #4:

Remarks to the Author:

Comments on the revision R1

The authors have improved the presentation of main results in terms of clarity and transparency. While I do not ride the enthusiasm of predictive deep neural networks in gene expression regulation in general, this is part of the advances that are inevitable to occur in the future, and the brain proteomics is an area where such advances should be appreciated the most given the obvious challenge in securing high quality tissue specimens during the disease incidence and progression.

While I do not intend to stay in the way for the authors to publish this article, I do however feel obliged to make a cautionary comment, to which the authors do not need to reply. The real bottleneck for me is the fact that the authors justify the modest prediction accuracy (at least in my judgment based on the data shown in Figures 2a, 2d) by the following argument: the proposed method performs better for the prediction of protein (Y) given mRNA (X) than other competing predictive models (cross-sample Pearson correlation against experimental data below 0.3 for most proteins), but when they correlate Y with clinical phenotype (Z), then they see more significant associations. Meanwhile, the abstract still says that the predictions of protein abundances are accurate. Though I have no objection to the downstream findings such as the pseudo-time analysis and the modular interpretation based on the *in silico* translated proteome data, these findings do not guarantee that the predicted accuracy of individual proteins in specific samples – as far as the predictive accuracy is concerned, the evidence suggests that the prediction is not accurate at all for many, many proteins.

This is a real danger in my opinion in terms of how these predicted protein data may be misinterpreted by readers, who will easily go beyond the point that the predicted data can be used for cohort-scale analyses. In fact, many deep learning enthusiasts are likely to neglect the fact that the accuracy (and precision) of the individual protein abundance values is subpar.

I'm not certain if *in silico* translation will ever be as good as global proteome analysis by experimental means because post-transcriptional regulation is complex, tissue specific, and context specific. Proteome-scale assays such as mass spectrometry or alternative technologies (e.g. aptamer-based assays) are becoming more accessible and economical year after year. Further, protein translation and degradation are regulated distinctly for each and every protein (e.g. kinases or scaffold proteins versus cytoskeletal proteins or cell surface glycoproteins) and protein expression regulation is often driven by different biological cues in major constituent cell populations of a tissue. Entrusting a network of perceptrons to capture these complex organizations sounds like a stretch, given my past experience with cellular dynamics analysis using both transcriptome and proteome data sets (PMID 30272558, 29263799, 26792871).

Reviewer #5:

Remarks to the Author:

The authors have addressed my concerns and I am ok with the manuscript in its current form.

Reviewer #6:

Remarks to the Author:

The authors have fully addressed all comments in the revised manuscript, which is greatly polished.

However, there is one more question: in the response to reviewers, the authors emphasize that "The difference between clei2block and NN-linear is that replacing the scGen module that executes non-linear transformation of input variables with a fully connected layer that operates linear transformation. This indicates that non-linear transformation is critical to achieving higher predictive accuracy". Just like in Fig. 2 b-c, the scGen performs better than NN-linear model; however, in Fig. 3a, removing LM model with mRNA dampens the accuracy the most. And authors mention that NN-linear is a combination of a fully connected layer and the LR module, implying that NN-linear model contains the information of the LR module. So why the NN-linear model is worse than the scGen only model. So which part is critical to the whole pipeline. More explanation is expected from the authors.

REVIEWER COMMENTS

Reviewer #4 (Remarks to the Author):

The authors have improved the presentation of main results in terms of clarity and transparency. While I do not ride the enthusiasm of predictive deep neural networks in gene expression regulation in general, this is part of the advances that are inevitable to occur in the future, and the brain proteomics is an area where such advances should be appreciated the most given the obvious challenge in securing high quality tissue specimens during the disease incidence and progression.

While I do not intend to stay in the way for the authors to publish this article, I do however feel obliged to make a cautionary comment, to which the authors do not need to reply. The real bottleneck for me is the fact that the authors justify the modest prediction accuracy (at least in my judgment based on the data shown in Figures 2a, 2d) by the following argument: the proposed method performs better for the prediction of protein (Y) given mRNA (X) than other competing predictive models (cross-sample Pearson correlation against experimental data below 0.3 for most proteins), but when they correlate Y with clinical phenotype (Z), then they see more significant associations. Meanwhile, the abstract still says that the predictions of protein abundances are accurate. Though I have no objection to the downstream findings such as the pseudo-time analysis and the modular interpretation based on the in silico translated proteome data, these findings do not guarantee that the predicted accuracy of individual proteins in specific samples – as far as the predictive accuracy is concerned, the evidence suggests that the prediction is not accurate at all for many, many proteins.

This is a real danger in my opinion in terms of how these predicted protein data may be misinterpreted by readers, who will easily go beyond the point that the predicted data can be used for cohort-scale analyses. In fact, many deep learning enthusiasts are likely to neglect the fact that the accuracy (and precision) of the individual protein abundance values is subpar.

I'm not certain if in silico translation will ever be as good as global proteome analysis by experimental means because post-transcriptional regulation is complex, tissue specific, and context specific. Proteome-scale assays such as mass spectrometry or alternative technologies (e.g. aptamer-based assays) are becoming more accessible and economical year after year. Further, protein translation and degradation are regulated distinctly for each and every protein (e.g. kinases or scaffold proteins versus cytoskeletal proteins or cell surface glycoproteins) and protein expression regulation is often driven by different biological cues in major constituent cell populations of a tissue. Entrusting a network of perceptrons to capture these complex organizations sounds like a stretch, given my past experience with cellular dynamics analysis using both transcriptome and proteome data sets (PMID 30272558, 29263799, 26792871).

Response: We appreciate your comments. We agree with the reviewer that the prediction won't go beyond the actual measurement. Our motivation for this work is to maximize the utility of existing and coming RNA-seq data for nominating disease-related genes that are likely to be altered at the protein level. We believe that the utility of our approach in disease protein discovery is demonstrated by the results mostly presented in Figure 4. In terms of the prediction

accuracy for protein levels rather than disease associations, it is close to the theoretical limit. However, as the reviewer pointed out, readers might expect a much higher correlation by the use of “accurate” in the abstract. Therefore, we revised the abstract to put the focus on disease associations. Thank you again for your thoughtful comment.

Reviewer #6 (Remarks to the Author):

The authors have fully addressed all comments in the revised manuscript, which is greatly polished.

However, there is one more question: in the response to reviewers, the authors emphasize that "The difference between clei2block and NN-linear is that replacing the scGen module that executes non-linear transformation of input variables with a fully connected layer that operates linear transformation. This indicates that non-linear transformation is critical to achieving higher predictive accuracy". Just like in Fig. 2 b-c, the scGen performs better than NN-linear model; however, in Fig. 3a, removing LM model with mRNA dampens the accuracy the most. And authors mention that NN-linear is a combination of a fully connected layer and the LR module, implying that NN-linear model contains the information of the LR module. So why the NN-linear model is worse than the scGen only model. So which part is critical to the whole pipeline. More explanation is expected from the authors.

Response: We apologize for the lack of clarity. The experiment in Fig. 3a focuses on gauging the usefulness of each data type in the prediction model by removing each data type. This evaluation quantifies the predictive power uniquely attributed to each data type, which is essentially a different test from the experiments in Fig. 2b-c. For instance, if there are two data types that have similar information for predicting protein abundance, the lack of one data type can be compensated by the other one and thus unique predictive contribution from these two data types is small. Fig. 3a indicates that mRNA has the most predictive signal that is distinct from the other data types.

In Fig. 2c, the scGen model alone gives us about 0.14 of correlation values that is better than mRNA alone (correlation is about 0.09). This indicates that although mRNA expression is a solo critical predictor in the model, the inputs for the scGen module collectively have a larger impact than mRNA alone. This seems to contradict Fig. 3a but again these two are not comparable as the metric in Fig. 3a is for evaluating the predictive signal unique to each data type. Fig. 2c indicates that the scGen contribution is overall important to achieve the higher performance and the NN-linear model cannot extract the meaningful information from the scGen inputs. Thus, the scGen part is critical in the clei2block architecture. We'd thank the reviewer to bring up this point. We revised the manuscript to clarify the aim of the experiment.